# Deep Bandits Show-Off:
# Simple and Efficient Exploration with Deep Networks

**Rong Zhu**
Institute of Science and Technology for Brain-inspired Intelligence, Fudan University
`rongzhu@fudan.edu.cn`

**Mattia Rigotti**[*]
IBM Research AI
`mr2666@columbia.edu`

## Abstract

Designing efficient exploration is central to Reinforcement Learning due to the fundamental problem posed by the exploration-exploitation dilemma. Bayesian exploration strategies like Thompson Sampling resolve this trade-off in a principled way by modeling and updating the distribution of the parameters of the action-value function, the outcome model of the environment. However, this technique becomes infeasible for complex environments due to the computational intractability of maintaining probability distributions over parameters of outcome models of corresponding complexity. Moreover, the approximation techniques introduced to mitigate this issue typically result in poor exploration-exploitation trade-offs, as observed in the case of deep neural network models with approximate posterior methods that have been shown to underperform in the deep bandit scenario.

In this paper we introduce *Sample Average Uncertainty (SAU)*, a simple and efficient uncertainty measure for contextual bandits. While Bayesian approaches like Thompson Sampling estimate outcomes uncertainty indirectly by first quantifying the variability over the parameters of the outcome model, SAU is a frequentist approach that directly estimates the uncertainty of the outcomes based on the value predictions. Importantly, we show theoretically that the uncertainty measure estimated by SAU asymptotically matches the uncertainty provided by Thompson Sampling, as well as its regret bounds. Because of its simplicity SAU can be seamlessly applied to deep contextual bandits as a very scalable drop-in replacement for epsilon-greedy exploration. We confirm empirically our theory by showing that SAU-based exploration outperforms current state-of-the-art deep Bayesian bandit methods on several real-world datasets at modest computation cost, and make the code to reproduce our results available at `https://github.com/ibm/sau-explore`.

## 1 Introduction

The *exploration-exploitation dilemma* is a fundamental problem in models of decision making under uncertainty in various areas of statistics, economics, machine learning, game theory, adaptive control and management. Given a set of actions associated with unknown probabilistic rewards, an agent has to decide whether to exploit familiar actions to maximizing immediate reward or to explore poorly understood or unknown actions for potentially finding ways to improve future rewards.

---

[*]Corresponding author

35th Conference on Neural Information Processing Systems (NeurIPS 2021).

Quantifying the uncertainty associated with the value of each action is a key component of conventional algorithms for addressing the exploration-exploitation dilemma. In particular, it is central to the two most successful exploration strategies commonly adopted in bandit settings: *Upper Confidence Bound* (UCB) and *Thompson Sampling*. The UCB algorithm [1–11] follows the principle of *optimism in the face of uncertainty*, which promotes exploration by maintaining confidence sets for action-value estimates and then choosing actions optimistically within these confidence sets. Thompson Sampling (TS), introduced by [12] and successfully applied in a wide range of settings [13–16], is based on the principle of *sampling in the face of uncertainty*, meaning that it samples actions from the posterior distribution over action-values given past rewards.

In modern reinforcement learning (RL), the flexible generalization capabilities of neural networks brought about by Deep RL have proven successful in tackling complex environments by learning mappings from high-dimensional observations directly to value estimates [17]. However, obtaining uncertainty measures over complex value functions like neural network models becomes challenging because of the intractability of estimating and updating posteriors over their parameters, limiting the applicability of Bayesian exploration strategies like UCB and TS. Recently, several proposals to address this challenge have been put forth that rely on approximations of the posterior over value functions. Unfortunately, these methods tend to underperform empirically compared to much simpler heuristics. For instance, [18] showed that in contextual bandit tasks the main approximate Bayesian posterior methods for deep neural networks are consistently beaten by simple baselines such as combining neural network value functions with a basic exploration strategy like epsilon-greedy, or using simple action-values like linear regression where the exact posterior can be computed.

In this paper we propose a novel uncertainty measure which departs from the Bayesian approach of estimating the uncertainty over the parameters of the value prediction model. Our uncertainty measure, which we call *Sample Average Uncertainty (SAU)* is a frequentist quantity that only depends on the value prediction of each action. In particular, unlike UCB and TS, exploration based on SAU does not require the costly computation of a posterior distribution over models in order to estimate uncertainty of their predictions. In fact, instead of first estimating the uncertainty over the parameters of the value function to then use it to quantify the uncertainty over outcomes, SAU directly estimates uncertainty over outcomes by measuring the variance of sample averages. This result is then plugged into the current estimate of the outcome model.

With our new measure of uncertainty of the expected action-values, we build two SAU-based exploration strategies: one based on the principle of *"optimism in the face of SAU"* that we name SAU-UCB, and a second one based on *"sampling in the face of SAU"* that we name SAU-Sampling.

We investigate the use of these new exploration strategies to tackle contextual bandit problems, and show that SAU is closely related to the mean-squared error in contextual bandits. This allows us to show analytically that in the case of Bernoulli multi-armed bandits the SAU measure converges to the uncertainty of the action-value estimates that are obtained by TS, despite SAU being much simpler to compute and not needing to rely on maintaining the posterior distribution. In addition, we derive an upper bound on the expected regret incurred by our SAU algorithms in multi-armed bandits that shows that they achieve the optimal logarithmic regret.

Finally, we empirically study the deployment of SAU-UCB and SAU-Sampling in the deep bandit setting and use them as exploration strategy for deep neural network value function models. Concretely, we follow the study of [18] and show that SAU consistently outranks the deep Bayesian bandit algorithms that they analyzed on the benchmarks that they proposed.

## 2   Problem Formulation: Contextual Bandits

The contextual bandit problem is a paradigmatic model for the study of the exploration-exploitation trade-off and is formulated as follows. At each time step $n$ we observe a context $\mathbf{x}_n$, select an action $a_n$ from a set $\mathbb{K} = \{1, \ldots, K\}$, after which we receive a reward $r_n$. The *value of an action* $a$ (in context $\mathbf{x}_n \in \mathbb{R}^p$) is defined as the expected reward given that $a$ is selected:

$$\mathbb{E}[r_n | a_n = a] = \mu(\mathbf{x}_n, \boldsymbol{\theta}_a), \tag{1}$$

where in general the action-values $\mu(\cdot)$ depend on unknown parameters $\boldsymbol{\theta}_a \in \mathbb{R}^p$.

Our goal is to design a sequential decision-making policy $\pi$ that over time learns the action parameters $\boldsymbol{\theta}_a$ which maximize the expected reward. This goal is readily quantified in terms of minimizing

*expected regret*, where we say that at step $n$ we incur expected regret

$$\max_{a' \in \mathbb{K}}\{\mu(\mathbf{x}_n, \boldsymbol{\theta}_{a'})\} - \mu(\mathbf{x}_n, \boldsymbol{\theta}_{a_n}), \tag{2}$$

i.e. the difference between the reward received by playing the optimal action and the one following the chosen action $a_n$. One way to design a sequential decision-making policy $\pi$ that minimizes expected regret is to quantify the uncertainty around the current estimate of the unknown parameters $\boldsymbol{\theta}_a$. TS for instance does this by sequentially updating the posterior of $\boldsymbol{\theta}_a$ after each action and reward. This paper presents a novel and simpler alternative method to estimate uncertainty.

## 3 Exploration based on Sample Average Uncertainty

### 3.1 Sample Average Uncertainty (SAU)

In this section, we begin with introducing our novel measure of uncertainty SAU. Let $\mathbb{T}_a$ denote the set of time steps when action $a$ was chosen so far, and let $n_a$ be the size of this set. Based on the $n_a$ rewards $\{r_n\}_{n \in \mathbb{T}_a}$ obtained with action $a$, the sample mean reward given action $a$ is:

$$\bar{r}_a = n_a^{-1} \sum\nolimits_{n \in \mathbb{T}_a} r_n.$$

At this point we reiterate that exploitation and exploration are customarily traded off against each other with a Bayesian approach that estimates the uncertainty of the action-values on the basis of a posterior distribution over their parameters given past rewards. Instead, we propose a *frequentist approach* that directly measures the uncertainty of the sample average rewards that was just computed. Direct calculation using eq. (1) then gives us that the variance of the sample mean reward is

$$\text{Var}(\bar{r}_a) = \bar{\sigma}_a^2/n_a, \quad \text{where} \quad \bar{\sigma}_a^2 = n_a^{-1} \sum\nolimits_{n \in \mathbb{T}_a} \sigma_{n,a}^2 \quad \text{with} \quad \sigma_{n,a}^2 = \mathbb{E}\left[(r_n - \mu(x_n, \boldsymbol{\theta}_a))^2\right].$$

Assuming that there is a sequence of estimators $\{\hat{\boldsymbol{\theta}}_{n,a}\}_{n \in \mathbb{T}_a}$ of $\boldsymbol{\theta}_a$, we can replace $\boldsymbol{\theta}_a$ with $\hat{\boldsymbol{\theta}}_{n,a}$ at each $n \in \mathbb{T}_a$ to approximate $\bar{\sigma}_a^2$ with a convenient statistics $\tau_a^2$ defined as

$$\tau_a^2 = n_a^{-1} \sum\nolimits_{n \in \mathbb{T}_a} \left(r_n - \mu(x_n, \hat{\boldsymbol{\theta}}_{n,a})\right)^2. \tag{3}$$

With this we get an approximate sample mean variance of

$$\widehat{\text{Var}}(\bar{r}_a) = \tau_a^2/n_a. \tag{4}$$

The central proposal of this paper is to use $\widehat{\text{Var}}(\bar{r}_a)$ as a measure of the uncertainty of the decision sequence. We call this quantity *Sample Average Uncertainty* (SAU), since it measures directly the uncertainty of sample mean rewards $\bar{r}_a$. In practice, $\tau_a^2$ can be updated incrementally as follows:

1. Compute the *prediction residual*: $\qquad\qquad e_n = r_n - \mu(\mathbf{x}_n, \hat{\boldsymbol{\theta}}_{n,a_n}); \tag{5}$

2. Update *Sample Average Uncertainty (SAU)*: $\quad \tau_{a_n}^2 \leftarrow \tau_{a_n}^2 + n_{a_n}^{-1}\left[e_n^2 - \tau_{a_n}^2\right]. \tag{6}$

Let us take a moment to contrast the uncertainty measure given by SAU and existing exploration algorithms like TS, which as we said would estimate the uncertainty of the action-value function $\mu(\cdot)$ by maintaining and updating a distribution over its parameters $\boldsymbol{\theta}_a$. SAU instead directly quantifies the uncertainty associated with each action by measuring the uncertainty of the sample average rewards. The clear advantage of SAU is that it is simple and efficient to compute: all it requires are the prediction residuals $r_n - \mu(\mathbf{x}_n, \hat{\boldsymbol{\theta}}_{n,a_n})$ without any need to model or access the uncertainty of $\mu(\mathbf{x}_n, \hat{\boldsymbol{\theta}}_{n,a})$. Because of the simplicity of its implementation, SAU can be naturally adapted to arbitrary action-value functions. In particular, it can be used to implement an exploration strategy for action-value function parameterized as deep neural networks or other model classes for which TS would be infeasible because of the intractability of computing a probability distribution over models.

Note that in updating $\tau_a^2$ we use the residuals obtained at each step rather than re-evaluating them using later estimates. This is a design choice motivated by the goal of minimizing the computation cost and implementation efficiency of SAU. Moreover, this choice can be justified from the viewpoint of the statistical efficiency, since, as the number of training samples increases, the impact of initial residuals

will decrease, so that the benefit of re-evaluating them incurs diminishing returns. Proposition 3 formalizes this argument by showing that indeed $\tau_a^2$ as computed in eq. (6) is concentrated around its expectation. In addition, perhaps as importantly, the aim of SAU is to provide a quantity to support exploration. The effect of potentially inaccurate residuals in the initial steps may actually be beneficial due to the introduction of additional noise driving initial exploration. This might be in part at the root of the good empirical results.

## 3.2 SAU-based Exploration in Bandit Problems

We now use the SAU measure to implement exploration strategies for (contextual) bandit problems.

**SAU-UCB.** UCB is a common way to perform exploration. Central to UCB is the specification of an "exploration bonus" which is typically chosen to be proportional to the measure of uncertainty. Accordingly, we propose to use the SAU measure $\tau_a^2$ as exploration bonus. Specifically, given value predictions $\hat{\mu}_{n,a} = \mu(\mathbf{x}_n, \hat{\boldsymbol{\theta}}_{n,a})$ for each $a$ at step $n$, we modify the values as

$$\widetilde{\mu}_{n,a} = \hat{\mu}_{n,a} + \sqrt{n_a^{-1}\tau_a^2 \log n}, \tag{7}$$

then choose the action by $a_n = \arg\max_a(\{\widetilde{\mu}_{n,a}\}_{a\in\mathbb{K}})$. We call this implementation of UCB using SAU as exploration bonus: SAU-UCB.

**SAU-Sampling.** "Sampling in the face of uncertainty" is an alternative exploration principle that we propose to implement with SAU in addition to UCB. This is inspired by TS which samples the success probability estimate $\hat{\mu}_a$ from its posterior distribution. Analogously, we propose to sample values from a parametric Gaussian distribution with a mean given by the value prediction and a variance given by $\bar{\sigma}_a^2$. This results in sampling values $\widetilde{\mu}_{n,a}$ at each time $n$ as:

$$\widetilde{\mu}_{n,a} \sim \mathcal{N}\left(\hat{\mu}_{n,a}, \tau_a^2/n_a\right), \tag{8}$$

then choosing the action by $a_n = \arg\max_a(\{\widetilde{\mu}_{n,a}\}_{a\in\mathbb{K}})$. We call this use of SAU inspired by TS, SAU-Sampling.

SAU-UCB and SAU-Sampling are summarized in Algorithm 1.

---

**Algorithm 1** SAU-UCB and SAU-Sampling for bandit problems

---

1: **Initialize:** $\hat{\boldsymbol{\theta}}_a$, $S_a^2 = 1$ and $n_a = 0$ for $a \in \mathbb{K}$.
2: **for** $n = 1, 2, \ldots$ **do**
3:     Observe context $\mathbf{x}_n$;
4:     **for** $a = 1, \ldots, K$ **do**
5:         Calculate the prediction $\hat{\mu}_{n,a} = \mu(\mathbf{x}_n; \hat{\boldsymbol{\theta}}_a)$ and $\tau_a^2 = S_a^2/n_a$;
6:         Draw a sample

$$\widetilde{\mu}_{n,a} = \hat{\mu}_{n,a} + \sqrt{\tau_a^2 n_a^{-1} \log n} \text{ (SAU-UCB)} \quad \text{or} \quad \widetilde{\mu}_{n,a} \sim \mathcal{N}\left(\hat{\mu}_{n,a}, n_a^{-1}\tau_a^2\right) \text{ (SAU-Sampling)};$$

7:     **end for**
8:     Compute $a_n = \arg\max_a(\{\widetilde{\mu}_{n,a}\}_{a\in\mathbb{K}})$ if $n > K$, otherwise $a_n = n$;
9:     Select action $a_n$, observe reward $r_n$;
10:    Update $\hat{\boldsymbol{\theta}}_{a_n}$ and increment $n_{a_n} \leftarrow n_{a_n} + 1$;
11:    Update $S_{a_n}^2 \leftarrow S_{a_n}^2 + e_n^2$ using prediction error calculated as $e_n = r_n - \hat{\mu}_{n,a_n}$;
12: **end for**

---

## 3.3 Novelty and comparison with related approaches

Using the variance estimation in MAB is not novel. For example [19] makes use of Bernstein's inequality to refine confidence intervals by additionally considering the uncertainty from estimating variance of reward noise. Our approach is fundamentally different from it with two aspects. First, Algorithm 1 is to propose a novel measure to approximate the uncertainty of the estimate of the mean reward that would afford such a flexible implementation and can therefore directly extended

and scaled up to complicated value models like deep neural networks. Second, our SAU quantity $\tau^2$ is the per-step squared prediction error, i.e., the average cumulative squared prediction error, as opposed to an estimate of the variance of the different arms. In fact, $\tau^2$ does not rely on the traditional variance estimation analyzed by[19], but is instead simply computed directly from the prediction. This difference makes SAU even easier to implement and adapt to settings like deep networks.

The exploration bonus in Algorithm 1 is not a function of the observed context, though it is updated from historical observations of the context. The algorithm could indeed be extended to provide a quantification of reward uncertainty that is a function of the current context by, for instance, fitting the SAU quantity as a function of context. Clearly, this will come at the cost of substantially increasing the complexity of the algorithm. Therefore to avoid this additional complexity, we instead focus the paper on the development of the SAU quantity as a simple estimate of uncertainty to efficiently drive exploration. However, exploring this possibility is a potentially exciting direction for future work.

# 4 SAU in Multi-Armed Bandits

## 4.1 SAU Approximates Mean-squared Error and TS in Multi-armed Bandits

Before considering the contextual bandits scenario, we analyze the measure of uncertainty provided by SAU in multi-armed bandits, and compare it to the uncertainty computed by TS. This will help motivate SAU and elucidate its functioning.

We assume a *multi-armed Bernoulli bandit*, i.e. at each step $n$ each action $a \in \mathbb{K}$ results in a reward sampled from $r_n \sim \text{Bernoulli}(\mu_a)$ with fixed (unknown) means $\mu_a \in [0, 1]$. Assume that action $a$ has been taken $n_a$ times so far, and let $\hat{\mu}_a$ denote the sample averages of the rewards for each action. The *prediction residual* eq. (5) is $e_n = r_n - \hat{\mu}_{a_n}$ and is the central quantity to compute SAU.

**TS in the case of Bernoulli bandits** is typically applied by assuming that the prior follows a Beta distribution, i.e. the values are sampled from $\text{Beta}(\alpha_a, \beta_a)$ with parameters $\alpha_a$ and $\beta_a$ for $a \in \mathbb{K}$. Uncertainty around the estimated mean values are then quantified by its variance denoted by $\hat{V}_a$ (see Appendix A.1). We then have the following proposition relating SAU and TS in Bernoulli bandits:

**Proposition 1** *For Beta Bernoulli bandits the expectation of the average* prediction residual $e_n^2/n_{a_n}$ *is an approximate unbiased estimator of the expectation of the* posterior variance $\hat{V}_a$ *in TS. Concretely:*

$$\mathbb{E}[\hat{V}_{a_n}] = \mathbb{E}[e_n^2/n_{a_n}] + O\left(n_{a_n}^{-2}\right).$$

**Proof** Proof of Proposition 1 is provided in Appendix A.1. ∎

Proposition 1 says that SAU asymptotically approximates TS for Bernoulli bandits, despite not needing to assume a prior and update a posterior distribution over parameters. In Appendix A.3 we support this empirically by showing that in multi-armed bandits SAU rivals TS.

The following proposition further characterizes the prediction residual:

**Proposition 2** *For Bernoulli bandits the expectation of the prediction residual used in SAU satisfies*

$$\mathbb{E}[e_n^2/n_{a_n}] = \mathbb{E}[(r_n - \hat{\mu}_{a_n})^2/n_{a_n}] = \mathbb{E}\left[(\hat{\mu}_{a_n} - \mu_{a_n})^2\right] + O\left(n_{a_n}^{-2}\right).$$

**Proof** Proof of Proposition 2 is provided in Appendix A.2. ∎

Proposition 2 says that the prediction residual $e_n = r_n - \hat{\mu}_{a_n}$ is an approximately unbiased estimator of the mean squared error $\mathbb{E}\left[(\hat{\mu}_{a_n} - \mu_{a_n})^2\right]$. This means that for Bernoulli bandits, SAU closely approximates the uncertainty of the action-value estimates.

Armed with this characterization of the prediction residual $r_n - \hat{\mu}_{a_n}$ in Proposition 2, we now quantify the performance of the estimator $\tau_a^2$ in eq. (3) in terms of its concentration around its expectation:

**Proposition 3** *For $\delta \in \left[2\exp\left(-\sigma_a^2 n_a/(32c)\right), 1\right)$, where $\sigma_a^2$ is the variance of $r_j$ for $j \in \mathbb{T}_a$ and $c$ a constant, we have*

$$\Pr\left\{\left|\tau_a^2 - \mathbb{E}\left[\tau_a^2\right]\right| \geq \sigma_a\sqrt{8c/(n_a\log(\delta/2))}\right\} \leq \delta,$$

**Proof** Proof of Proposition 3 is provided in Appendix A.4. ∎

Proposition 3 says that $\tau_a^2$ is concentrated around its expectation, and thus remains stable as it is being updated. In Appendix A.6 we also show that $\mathbb{E}\left[\tau_a^2\right] \to \sigma_a^2$ as $n_a \to \infty$, and in Appendix A.7 we derive an upper bound on the expected regrets of SAU-UCB and SAU-Sampling in multi-armed bandits proving that the optimal logarithmic regrets are achievable uniformly over time, which says that the theoretical performance of SAU rivals TS in multi-armed bandits.

## 4.2 SAU in Linear Contextual Bandits: Theoretical analysis

We now show that the results in Proposition 2 also hold for another important bandit model beside Bernoulli bandits, i.e. *linear contextual bandits* defined by the following outcome model:

$$r_n = \mathbf{x}_n^\top \boldsymbol{\theta}_a + \epsilon_{n,a}, \quad n = 1, 2, \ldots, \tag{9}$$

where $\mathbf{x}_n, \boldsymbol{\theta}_a \in \mathbb{R}^p$, and $\epsilon_{n,a}$ are iid random variables with variance $\sigma_a^2$. Assume action $a$ was selected $n_a$ times. We obtain the least-squares estimator $\hat{\boldsymbol{\theta}}_{n,a_n} = (\sum_{j \in \mathbb{T}_{n,a_n}} \mathbf{x}_j^\top \mathbf{x}_j)^{-1}(\sum_{j \in \mathbb{T}_{n,a_n}} \mathbf{x}_j^\top r_j)$. Accordingly, the prediction and the prediction residual at step $n$ are, respectively,

$$\hat{\mu}_{n,a_n} = \mathbf{x}_n^\top \hat{\boldsymbol{\theta}}_{n,a_n} \quad \text{and} \quad e_n^2 = (r_n - \mathbf{x}_n^\top \hat{\boldsymbol{\theta}}_{n,a_n})^2. \tag{10}$$

Denote $h_n = \mathbf{x}_n^\top (\sum_{j \in \mathbb{T}_{n,a_n}} \mathbf{x}_j^\top \mathbf{x}_j)^{-1} \mathbf{x}_n$. The mean squared error of $\mathbf{x}_n^\top \hat{\boldsymbol{\theta}}_{n,a_n}$ is $\text{MSE}_n = \mathbb{E}[(\mathbf{x}_n^\top \hat{\boldsymbol{\theta}}_{n,a_n} - \mathbf{x}_n^\top \boldsymbol{\theta}_{a_n})^2]$. With direct calculation we see that $\text{MSE}_n = h_n \sigma_{a_n}^2$ and that $\mathbb{E}\left[e_n^2 / n_{a_n}\right] = (1 - h_n)\sigma_{a_n}^2 / n_{a_n}$. Therefore, we have the following proposition:

**Proposition 4** *For linear contextual bandits* (9) *we have that*

$$\mathbb{E}[e_n^2 / n_{a_n}] = (h_n n_{a_n})^{-1}(1 - h_n)\,\text{MSE}_n.$$

*Furthermore, assuming that there exist constants $c_1$ and $c_2$ so that $c_1/n_{a_n} \leq h_n \leq c_2/n_{a_n}$, then*

$$c_2^{-1}(1 - c_2/n_{a_n})\,\text{MSE}_n \leq \mathbb{E}\left[e_n^2 / n_{a_n}\right] \leq c_1^{-1}(1 - c_1/n_{a_n})\,\text{MSE}_n.$$

Proposition 4 provides a lower and an upper bound for $\mathbb{E}\left[e_n^2 / n_{a_n}\right]$ in terms of $\text{MSE}_n$, meaning that on average SAU is a conservative measure of the uncertainty around $\mathbf{x}_n^\top \hat{\boldsymbol{\theta}}_{n,a_n}$. Noting that $0 \leq h_j \leq 1$ and $\sum_{j \in \mathbb{T}_{n,a_n}} h_j = p$, the assumption that $c_1/n_{a_n} \leq h_n \leq c_2/n_{a_n}$ requires that $h_n$ does not dominate or is dominated by other terms $h_j$, with $j \in \mathbb{T}_{n,a_n}$, meaning that contexts should be "homogeneous" to a certain extent. To examine the robustness to violations of this assumption, in the simulation in Appendix B we empirically test the performance under a heavy-tailed $t$-distribution with $df = 2$. The results show that SAU works robustly even under such type of context inhomogeneity.

## 4.3 SAU in Linear Contextual Bandits: Empirical evaluation on synthetic data

In this section, we present simulation results quantifying the performance of our SAU-based exploration algorithms in linear contextual bandits. We evaluate SAU on synthetically generated datasets to address two questions: (1) How does SAU's performance compare against Thompson Sampling?, and (2) How robust is SAU in various parameter regimes?

We consider three scenarios for $K$ (the number of actions) and $p$ (the context dimensionality): (a) $K = 5, p = 5$, (b) $K = 20, p = 5$, and (b) $K = 5, p = 40$. The horizon is $N = 20000$ steps. For each action $a$, parameters $\boldsymbol{\theta}_a$ are drawn from a uniform distribution in $[-1, 1]$, then normalized so that $\|\boldsymbol{\theta}_a\| = 1$. Next, at each step $n$ context $\mathbf{x}_n$ is sampled from a Gaussian distribution $\mathcal{N}(\mathbf{0}_p, \mathbf{I}_p)$. Finally, we set the noise variance to be $\sigma^2 = 0.5^2$ so that the signal-to-noise ratio equals 4.

We compare our SAU-based exploration algorithms, SAU-UCB and SAU-Sampling to Thompson Sampling ("TS" in Fig. 1). For TS on linear model, we follow [18] and use Bayesian linear regression for exact posterior inference. We also consider the *PrecisionDiag* approximation for the posterior covariance matrix of $\boldsymbol{\theta}_a$ with the same priors as in [18] ("TSdiag" in Fig. 1).

Fig. 1a) shows regret as a function of step for $(K, p) = (5, 5)$. From the figure we have two observations: SAU-Sampling is comparable to TS, and SAU-UCB achieves better regret than TS. In

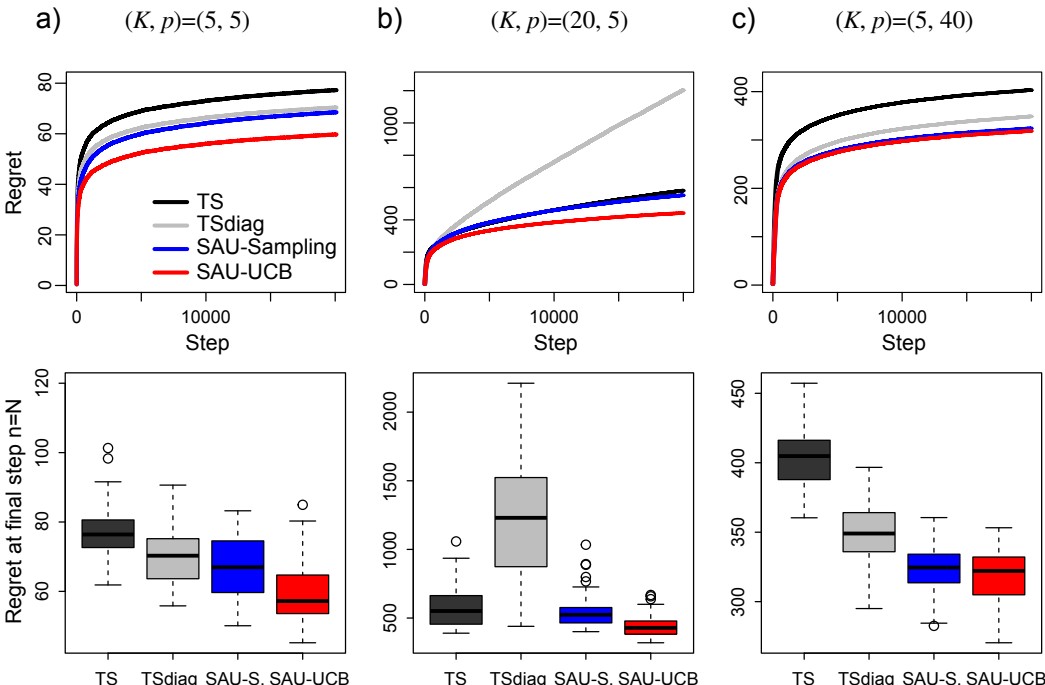

Figure 1: Performance on contextual linear bandits with various $(K, p)$ parameters showing that our models (SAU-Sampling and SAU-UCB) consistently achieve lower regret than TS using Bayesian linear regression with exact posterior inference (TS) and TS with *PrecisionDiag* approximation (TSdiag). The upper panels report regret as a function of step $n$, where results are averaged over 100 runs. The lower panels show the distributions of the regret at the final step.

terms of cumulative regret SAU significantly outperforms TS and TSdiag. Figures 1b) and c) show the effects of larger $K$ and $p$, respectively. The observations from Fig. 1a) still hold in these cases, implying that SAU's performance is robust to an increase in action space and context dimension.

We also consider four other cases: (1) the elements of $\boldsymbol{\theta}_a$ are sampled from $\mathcal{N}(0, 1)$ then are normalized; (2) the model errors are correlated with $AR(1)$ covariance structure with correlation $\rho = 0.5$; (3) the elements in $\mathbf{x}_i$ are correlated with $AR(1)$ covariance structure with correlation $\rho = 0.5$; and (4) the elements of $\mathbf{x}_i$ are sampled from a heavy-tailed $t$-distribution with $df = 2$ and are truncated at 5. These results are shown in Appendix B and are consistent with the results in Fig. 1 confirming SAU's robustness to various contextual linear bandit problems.

## 5 Deep Contextual Bandits

### 5.1 Deep Bayesian Bandit Algorithms

Deep contextual bandits refers to tackling contextual bandits by parameterizing the action-value function as a deep neural network $\mu(\mathbf{x}, \boldsymbol{\theta})$, thereby leveraging models that have been very successful in the large-scale supervised learning [20] and RL [17]. Notice that in the deep setting we denote all parameters with $\boldsymbol{\theta} = \{\boldsymbol{\theta}_a\}_{a \in \mathbb{K}}$, as common in the neural network literature. In particular, $\boldsymbol{\theta}$ includes the parameters that are shared across actions, as well as those of the last layer of the network which are specific to each action $a$. Algorithm 2 breaks down a generic deep contextual bandit algorithm in terms of an API exposing its basic subroutines: PREDICT (which outputs the set of action-values $\{\mu_{n,a}\}_{a \in \mathbb{K}}$ given the observation $\mathbf{x}_n$), ACTION (which selects an action given all the action-values), and UPDATE (which updates model parameters at the and of the step).

In this scheme Thompson Sampling (TS) is implemented as in Algorithm 3, which underlines where TS promotes exploration by sampling from a distribution over model parameters $P_n(\boldsymbol{\theta})$. In principle this provides an elegant Bayesian approach to tackle the exploration-exploitation dilemma embodied

---
**Algorithm 2** Generic Deep Contextual Bandit algorithm
---
1: **for** $n = 1, 2, \ldots$ **do**
2:      Observe context $\mathbf{x}_n$;
3:      Compute values $\{\mu_{n,a}\}_{a \in \mathbb{K}} = \text{PREDICT}(\mathbf{x}_n)$;
4:      Choose $a_n = \text{ACTION}(\{\mu_{n,a}\}_{a \in \mathbb{K}})$, observe reward $r_n$;
5:      UPDATE $(r_n, a_n, \mathbf{x}_n)$;
6: **end for**
---

by contextual bandits. Unfortunately, representing and updating a posterior distribution over model parameters $P_n(\boldsymbol{\theta})$ exactly becomes intractable for complex models such as deep neural networks.

---
**Algorithm 3** Thompson Sampling for Deep Contextual Bandits
---
1: **function** PREDICT($\mathbf{x}_n$)
2:      **Exploration:** Sample model parameters from posterior distribution: $\hat{\boldsymbol{\theta}}_n \sim P_n(\boldsymbol{\theta})$;
3:      **Return** predicted values $\{\hat{\mu}_{n,a}\}_{a \in \mathbb{K}} = \mu(\mathbf{x}_n, \hat{\boldsymbol{\theta}}_n)$, where
4: **function** ACTION($\{\hat{\mu}_{n,a}\}_{a \in \mathbb{K}}$)
5:      **Return** $a_n = \arg\max_a(\{\tilde{\mu}_{n,a}\}_{a \in \mathbb{K}})$;
6: **function** UPDATE($r_n, a_n, \mathbf{x}_n$)
7:      Use triplet $(r_n, a_n, \mathbf{x}_n)$ to update posterior distribution: $P_{n+1}(\boldsymbol{\theta}) \leftarrow P_n(\boldsymbol{\theta})$;
---

To obviate this problem, several techniques that heuristically approximate posterior sampling have emerged, such as randomly perturbing network parameters [21–23], or bootstrapped sampling [24]. Within the scheme of Algorithm 2 the role of random perturbation and bootstrapped sampling are to heuristically emulate the model sampling procedure promoting exploration in the PREDICT subroutine (see TS Algorithm 3). However, systematic empirical comparisons recently demonstrated that simple strategies such as epsilon-greedy [17, 25] and Bayesian linear regression [26] remain very competitive compared to these approximate posterior sampling methods in deep contextual bandit. In particular, [18] showed that linear models where the posterior can be computed exactly, and epsilon-greedy action selection overwhelmingly outrank deep methods with approximate posterior sampling in a suite of contextual bandit benchmarks based on real-world data.

## 5.2 SAU for Deep Contextual Bandits

We now re-examine the deep contextual bandits benchmarks in [18] and show that SAU can be seamlessly combined with deep neural networks, resulting in an exploration strategy whose performance is competitive with the best deep contextual bandit algorithms identified by [18].

Algorithm 4 shows the deep contextual bandit implementation of SAU. Notice that the PREDICT subroutine is remarkably simple, consisting merely in the forward step of the deep neural network value prediction model. In contrast to our extremely simple procedure, TS-based methods require at this step to (approximately) sample from the model posterior to implement exploration. In SAU exploration is instead taken care of by the ACTION subroutine, which takes the values as inputs and either explores through sampling from a distribution around the predicted values (SAU-Sampling) or through an exploration bonus added to them (SAU-UCB). SAU then selects the action corresponding to the maximum of these perturbed values. The UPDATE for SAU is also quite simple, and consists in updating the neural network parameters to minimize the reward prediction error loss $l_n$ following action selection using SGD via backprop, or possibly its mini-batch version (which would then be carried out on a batch of $(r_n, a_n, \mathbf{x}_n)$ triplets previously stored in a memory buffer). UPDATE then updates the count and the SAU measure $\tau_{a_n}$ for the selected action $a_n$.

We notice that the simplicity of SAU for deep contextual bandits is akin to the simplicity of epsilon-greedy, for which exploration is also implemented in the ACTION subroutine (see Algorithms 5 in Appendix E). In fact, comparing the two algorithms it is clear that SAU can be used as a *drop-in replacement for epsilon-greedy exploration*, making it widely applicable.

**Algorithm 4** SAU for Deep Contextual Bandits (SAU-Neural-Sampling and UCB)

1: **function** PREDICT($\mathbf{x}_n$)
2:     **Return** predicted values $\{\hat{\mu}_{n,a}\}_{a\in\mathbb{K}} = \mu(\mathbf{x}_n, \hat{\boldsymbol{\theta}}_n)$;

3: **function** ACTION($\{\hat{\mu}_{n,a}\}_{a\in\mathbb{K}}$)
4:     **Exploration:** Compute   $\boxed{\tilde{\mu}_{n,a} \sim \mathcal{N}\left(\hat{\mu}_{n,a}, \tau_a^2/n_a\right)}$    (SAU-Sampling)
5:                    or  $\boxed{\tilde{\mu}_{n,a} = \hat{\mu}_{n,a} + \sqrt{\tau_a \log n/n_a}}$   (SAU-UCB);
6:     **Return** $a_n = \arg\max_a(\{\tilde{\mu}_{n,a}\}_{a\in\mathbb{K}})$;

7: **function** UPDATE($r_n, a_n, \mathbf{x}_n$)
8:     Compute prediction error $e_n = r_n - \hat{\mu}_{n,a_n}$ and loss $l_n = \frac{1}{2}(r_n - \hat{\mu}_{n,a_n})^2$
9:     Update model parameters to $\hat{\boldsymbol{\theta}}_{n+1}$ using SGD with gradients $\frac{\partial l_n}{\partial\boldsymbol{\theta}}$ (or mini-batch version);
10:     Update exploration parameters: $n_{a_n} \leftarrow n_{a_n} + 1$,    $S_{a_n}^2 \leftarrow S_{a_n}^2 + e_n^2$    $\tau_{a_n}^2 = S_{a_n}^2/n_{a_n}$;

### 5.3 Empirical Evaluation of SAU on Deep Contextual Bandit Problems

**Benchmarks and baseline algorithms.** Our empirical evaluation of SAU's performance in the deep contextual bandit setting is based on the experiments by [18], who benchmarked the main TS-based approximate posterior sampling methods over a series of contextual bandit problems. We test SAU on the same contextual bandit problems against 4 competing algorithms consisting in the 4 best ranking algorithms identified by [18], which are: *LinearPosterior* (a closed-form Bayesian linear regression algorithm for exact posterior inference under the assumption of a linear contextual bandit [27]), *LinearGreedy* (epsilon-greedy exploration under the assumption of a linear contextual bandit), *NeuralLinear* (Bayesian linear regression on top of the last layer of a neural network trained with SGD [28]) and *NeuralGreedy* (a neural network with epsilon-greedy exploration trained with SGD). We neglected a direct comparison with NeuralUCB [29], since its scaling in memory and computational requirements make it quickly impractical for even moderately sized applications of practical interest. Moreover, its reported performance is substantially worse than SAU-UCB.

**Implementations of SAU.** We implemented and tested 4 versions of SAU on the benchmarks in [18]. In the Tables below we refer to them a follows: *Linear-SAU-S* and *Linear-SAU-UCB* refer to a linear regression model using SAU-Sampling and SAU-UCB as exploration strategies, respectively. *Neural-SAU-S* and *Neural-SAU-UCB* refer to a neural network model trained with SGD using SAU-Sampling and SAU-UCB, respectively.

**Empirical evaluation on the Wheel Bandit.** The *Wheel Bandit Problem* is a synthetic bandit designed by [18] to study the performance of bandit algorithms as a function of the need for exploration in the environment by varying a parameter $\delta \in [0, 1]$ that smoothly changes the importance of exploration. In particular, the difficulty of the problem increases with $\delta$, since the problem is designed so that for $\delta$ close to 1 most contexts have the same optimal action, while only for a fraction $1 - \delta^2$ of contexts the optimal action is a different more rewarding action (see [18] for more details). In Appendix C, Table 2 quantifies the performance of SAU-Sampling and SAU-UCB in terms of cumulative regret in comparison to the 4 competing algorithms, and normalized to the performance of the *Uniform* baseline, which selects actions uniformly at random. There we can see that *Neural-SAU-S* is consistently the best algorithm with lower cumulative regret for a wide rage of the parameter $\delta$. Only for very high values of $\delta$ ($\delta = 0.99$) the baseline algorithm *NeuralLiner* starts to overtake it, but even in this case, another variant of SAU, *SAU-Linear-S* still maintains the lead in performance.

**Empirical evaluation on real-world Deep Contextual Bandit problems.** Table 1 quantifies the performance of SAU-Sampling and SAU-UCB in comparison to the 4 competing baseline algorithms, and normalized to the performance of the *Uniform* baseline. These results show that a SAU algorithm is the best algorithm in each of the 7 benchmarks in terms of minimizing cumulative regret over all samples. *Neural-SAU-S* or *Neural-SAU-UCB* are the best combination 6 out of 7 times, and linear regression with SAU-UCB is the best on the bandit built from the Adult dataset. The next best algorithm in terms of minimizing cumulative regret is *NeuralLinear* [18], which incurs cumulative regret that on average is 32% higher than *Neural-SAU-S* and 34% higher than *Neural-SAU-UCB*.

As already mentioned, thanks to their implementation efficiency SAU-based algorithms are much less computation intensive than TS-based algorithms. This is reflected in the remarkably shorter

Table 1: Cumulative regret incurred on the contextual bandits in [18] by the 4 best algorithms that they identified (described in Appendix D) compared against our SAU-based algorithms. Results are relative to the cumulative regret of the Uniform algorithm. We report the mean and standard error of the mean over 50 trials. The best mean cumulative regret value for each task is marked in bold.

| | Mushroom | Statlog | Covertype | Financial | Jester | Adult | Census |
|---|---|---|---|---|---|---|---|
| LinearPosterior | 3.02 ± 0.15 | 10.29 ± 0.19 | 36.88 ± 0.07 | 10.77 ± 0.15 | 64.01 ± 0.40 | 75.87 ± 0.06 | 46.70 ± 0.08 |
| LinearGreedy | 4.64 ± 0.35 | 109.26 ± 0.95 | 53.50 ± 2.00 | 15.48 ± 1.02 | 62.71 ± 0.45 | 86.70 ± 0.24 | 65.35 ± 1.57 |
| NeuralLinear | 2.66 ± 0.08 | 1.26 ± 0.03 | 29.18 ± 0.07 | 10.16 ± 0.18 | 69.48 ± 0.43 | 78.28 ± 0.07 | 41.00 ± 0.09 |
| NeuralGreedy | 26.66 ± 0.32 | 40.17 ± 0.46 | 88.85 ± 0.12 | 85.85 ± 1.37 | 93.15 ± 0.77 | 98.96 ± 0.03 | 85.82 ± 0.26 |
| **Linear-SAU-S** | 4.58 ± 0.39 | 10.44 ± 0.26 | 36.29 ± 0.12 | 8.71 ± 0.52 | 62.59 ± 0.43 | 74.70 ± 0.13 | 39.65 ± 0.11 |
| **Linear-SAU-UCB** | 3.09 ± 0.15 | 10.02 ± 0.22 | 36.77 ± 0.17 | 6.51 ± 0.37 | 64.43 ± 0.40 | **74.62 ± 0.07** | 39.98 ± 0.09 |
| **Neural-SAU-S** | **2.20 ± 0.05** | 0.62 ± 0.01 | **27.46 ± 0.06** | 5.60 ± 0.11 | **61.02 ± 0.57** | 78.02 ± 0.07 | 38.76 ± 0.08 |
| **Neural-SAU-UCB** | 2.32 ± 0.06 | **0.60 ± 0.01** | 27.90 ± 0.06 | **5.26 ± 0.15** | 62.27 ± 0.61 | 78.09 ± 0.06 | **38.54 ± 0.09** |
| Uniform | 100.00 ± 0.24 | 100.00 ± 0.03 | 100.00 ± 0.02 | 100.00 ± 1.47 | 100.00 ± 0.96 | 100.00 ± 0.02 | 100.00 ± 0.04 |

execution time: on average *Neural-SAU-S* and *Neural-SAU-UCB* run more than 10 time faster than *NeuralLinear* [18] (see Appendix Table 5 for details), also making them extremely scalable.

## 6 Conclusion and Discussion

Existing methods to estimate uncertainty tend to be impractical for complex value function models like deep neural networks, either because exact posterior estimation become unfeasible, or due to how approximate algorithms coupled with deep learning training amplify estimation errors.

In this paper, we have introduced Sample Average Uncertainty (SAU), a simple and efficient uncertainty measure for contextual bandit problems which sidesteps the mentioned problems plaguing Bayesian posterior methods. SAU only depends on the value prediction, in contrast to methods based on Thompson Sampling that instead require an estimate of the variability of the model parameters. As a result, SAU is immune to the negative effects that neural network parameterizations and optimization have on the quality of uncertainty estimation, resulting in reliable and robust exploration as demonstrated by our empirical studies. SAU's implementation simplicity also makes it suitable as a drop-in replacement for epsilon-greedy action selection, resulting in a scalable exploration strategy that can be effortlessly deployed in large-scale and online contextual bandit scenarios.

We also have provided theoretical justifications for SAU-based exploration by connecting SAU with posterior variance and mean-squared error estimation. However, the reasons why SAU is in practice consistently better than TS-based exploration in deep bandits is still not settled theoretically. We hypothesize that this might be due to two main reasons: (1) TS-based methods implement exploration by estimating the uncertainty of the internal model parameters, which might introduce estimation errors, while SAU directly estimates uncertainty at the model output; (2) in addition, the approximation error from the approximate posterior implementations of TS-based models might result in inefficient uncertainty measures of the internal model parameters. Because of the importance of contextual bandit algorithms for practical applications like for instance recommendation and ad servicing systems, we believe that it will be important to further theoretically refine these hypotheses to help mitigate the possible negative societal impacts that could result from deploying inefficient, miscalibrated or biased exploration algorithms.

Another limitation of our work is that it developed SAU-based exploration in the specific and restricted case of bandits. Despite being an application of interest, we are excited and looking forward to further development that could extend methods based on SAU to more general sequential decision scenarios in RL beyond the bandit setting.

## Acknowledgements

This work was partially supported by National Natural Science Foundation of China (No.11871459) and by Shanghai Municipal Science and Technology Major Project (No.2018SHZDZX01).

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
