## Appendix A    Multi-armed bandits

### A.1    Thompson Sampling for Beta-Bernoulli bandits

Thompson Sampling (TS) in the case of Bernoulli bandits typically assumes that the true values $\mu_a$ are sampled from a Beta distribution with parameters $\alpha_a$ and $\beta_a$. Accordingly, TS samples the values from $\text{Beta}(\alpha_a, \beta_a)$. This distribution has mean $\hat{\mu}_a = \alpha_a/(\alpha_a + \beta_a)$ and variance $\hat{V}_a = (\alpha_a + \beta_a)^{-2}(\alpha_a + \beta_a + 1)^{-1}\alpha_a\beta_a$. The mean $\hat{\mu}_a$ represents exploitation, while the posterior variance $\hat{V}_a$ promotes exploration.

After an action $a$ is selected at step $n$, the posterior over $\mu_a$ is updated by updating the corresponding parameters as $(\alpha_a, \beta_a) \leftarrow (\alpha_a, \beta_a) + (r_n, 1 - r_n)$.

Direct calculation gives that

$$\mathbb{E}[\hat{V}_a] = \mathbb{E}[\hat{\mu}_a(1 - \hat{\mu}_a)/(n_a + 1)] = \mu_a(1 - \mu_a)/n_a + O\left(n_a^{-2}\right).$$

Comparing this last expression with eq. (11) proves Proposition 1.

### A.2    Proof of Proposition 2

We first note that:

$$\begin{aligned}
\mathbb{E}[e_n^2/n_{a_n}] &= \mathbb{E}\left[(r_n - \hat{\mu}_{a_n})^2/n_{a_n}\right] \\
&= \mathbb{E}\left[(r_n - \mu_{a_n} + \mu_{a_n} - \hat{\mu}_{a_n})^2\right]/n_a = \mu_{a_n}(1 - \mu_{a_n})/n_{a_n} + O\left(n_{a_n}^{-2}\right),
\end{aligned} \quad (11)$$

where the last step follows by noticing that $\mathbb{E}\left[(r_n - \mu_{a_n})^2\right] = \mu_{a_n}(1 - \mu_{a_n})$.

Recalling that $\hat{\mu}_a$ are sample averages of Bernoulli variables with means $\mu_a$, we note that

$$\mathbb{E}\left[(\hat{\mu}_a - \mu_a)^2\right] = \mu_a(1 - \mu_a)/n_a.$$

Inserting this result for $a = a_n$ in the previous expression results in

$$\mathbb{E}[e_n^2]/n_{a_n} = \mathbb{E}\left[(r_n - \hat{\mu}_{a_n})^2\right]/n_{a_n} = \mathbb{E}\left[(\hat{\mu}_{a_n} - \mu_{a_n})^2\right] + O\left(n_{a_n}^{-2}\right).$$

proving Proposition 2.

### A.3    Empirical Studies on Multi-armed bandits

We present a simple synthetic example that simulates Bernoulli rewards and investigates its performance. The reward distribution of action $a \in \{1, \ldots, K\}$, $P_a = \text{Bernoulli}(\mu_a)$, is parameterized by the expected reward $\mu_a \in [0, 1]$. In this simulation, the optimal action has a reward probability of $\mu_1$ and the $K - 1$ other actions have a probability of $\mu_1 - \epsilon$. We consider $\mu_1 = 0.5$, $\epsilon \in \{0.1, 0.02\}$ and $K \in \{10, 50\}$. The horizon is $n = 10^5$ rounds. Our SAU-based exploration approaches, SAU-UCB and SAU-Sampling, are compared to UCB1, and Thompson Sampling (TS).

Fig. 2a shows the regret as a function of round in the case $\epsilon = 0.1$ and $K = 10$. From the figure, we have two observations: (1) SAU-Sampling works similarly as TS; (2) SAU-UCB achieves a much smaller regret than UCB1, even work more or less than TS. This empirical performance shows the potential of our SAU measure.

We continue to empirically investigate the effects of $K$ and $\epsilon$ (Fig. 2c), and show the performance in non-Bernoulli payoff (Fig. 2d). We consider a larger number of actions $K = 50$ (Fig. 2b) and a smaller $\epsilon = 0.02$ (Fig. 2c). From these figures, the performance of $K = 50$ or $\epsilon = 0.02$ is consistent with the case $(K, \epsilon) = (10, 0.1)$. We also run a non-Bernoulli bandits, uniform bandit, in which a random variable $u$ is from uniform distribution $[0, 1]$, then the binary reward $r = 1$ if $u \leq \mu_a$, 0 otherwise. We plot the result in Fig. 2d, where the results works consistently as Bernoulli bandits. These results show that SAU works robustly to various multi-armed bandit problems.

### A.4    Proofs of bound Proposition 3

Let $\mathbf{r}_{n_a}$ be the sub-vector of $(r_1, r_2, \ldots)^\top$ corresponding to the indices in $\mathbb{T}_a$, and $\mathbf{1}_j = (1, \cdots, 1, 0, \cdots, 0)^\top$ a vector whose first $j$ components are 1 and all the others are zero. Denote

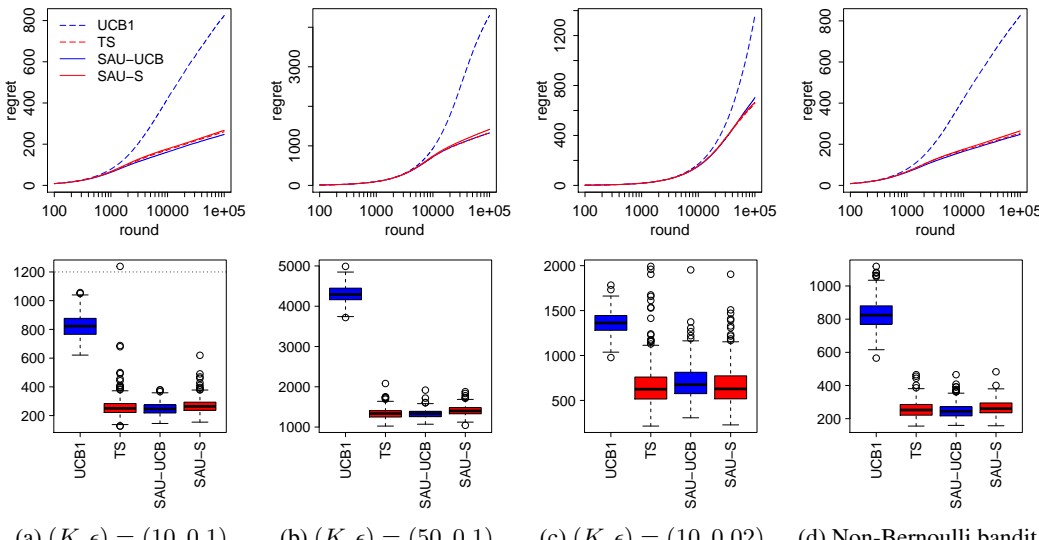

(a) $(K, \epsilon) = (10, 0.1)$.    (b) $(K, \epsilon) = (50, 0.1)$.    (c) $(K, \epsilon) = (10, 0.02)$.    (d) Non-Bernoulli bandit.

Figure 2: Performance on 4 multi-armed bandit problems: three Bernoulli bandits (a,b,c) and one non-Bernoulli (uniform) bandit (d). Upper panels report regret performance as a function of play $n$ averaged over 500 runs. Lower panels report regret distribution at the last play. For Bernoulli bandits rewards are sampled from Bernoulli($\mu_a$), best action has a reward probability of $\mu$ and the $K-1$ other actions have a probability of $\mu - \epsilon$. For the uniform bandit the rewards are sampled from $\mathbb{1}\{U[0,1] < \mu_a\}$, where $U[0,1]$ is a random variable from uniform distribution in $[0,1]$, $K = 10$ and $\epsilon = 0.1$.

$\mathbf{Q}_{n_a} = (\mathbf{1}_1, 2^{-1}\mathbf{1}_2, \cdots, n_a^{-1}\mathbf{1}_{n_a})^\top$. Eqn. (3) then becomes

$$\tau_a^2 = n_a^{-1} \sum_{j \in \mathbb{T}_a} [r_j - j^{-1}\mathbf{1}_j^\top \mathbf{r}_{n_a}]^2 = n_a^{-1} \mathbf{r}_{n_a}^\top (\mathbf{I}_{n_a} - \mathbf{Q}_{n_a})^\top (\mathbf{I}_{n_a} - \mathbf{Q}_{n_a}) \mathbf{r}_{n_a}.$$

Noting that $(\mathbf{I}_{n_a} - \mathbf{Q}_{n_a})\mathbf{1}_{n_a} = (0, 0, \cdots, 0)^\top$, we can write

$$\begin{aligned}
\tau_a^2 =& \frac{1}{n_a} \mathbf{r}_{n_a}^\top (\mathbf{I}_{n_a} - \mathbf{Q}_{n_a})^\top (\mathbf{I}_{n_a} - \mathbf{Q}_{n_a}) \mathbf{r}_{n_a} \\
=& \frac{1}{n_a} (\mathbf{r}_{n_a} - \mu_a \mathbf{1}_{n_a})^\top (\mathbf{I}_{n_a} - \mathbf{Q}_{n_a})^\top (\mathbf{I}_{n_a} - \mathbf{Q}_{n_a}) (\mathbf{r}_{n_a} - \mu_a \mathbf{1}_{n_a}) \\
& + \frac{2\mu_a}{n_a} (\mathbf{r}_{n_a} - \mu_a \mathbf{1}_{n_a})^\top (\mathbf{I}_{n_a} - \mathbf{Q}_{n_a})^\top (\mathbf{I}_{n_a} - \mathbf{Q}_{n_a}) \mathbf{1}_{n_a} - \frac{\mu_a^2}{n_a} \mathbf{1}_{n_a}^\top (\mathbf{I}_{n_a} - \mathbf{Q}_{n_a})^\top (\mathbf{I}_{n_a} - \mathbf{Q}_{n_a}) \mathbf{1}_{n_a} \\
=& \frac{1}{n_a} (\mathbf{r}_{n_a} - \mu_a \mathbf{1}_{n_a})^\top (\mathbf{I}_{n_a} - \mathbf{Q}_{n_a})^\top (\mathbf{I}_{n_a} - \mathbf{Q}_{n_a}) (\mathbf{r}_{n_a} - \mu_a \mathbf{1}_{n_a}) \\
=& : \tau_{a;e}^2.
\end{aligned} \tag{12}$$

Denote $\mathbf{P}_{n_a} = (\mathbf{I}_{n_a} - \mathbf{Q}_{n_a})^\top (\mathbf{I}_{n_a} - \mathbf{Q}_{n_a})$. By applying Theorem 2.5 in [30](Lemma 2 below), we have that for $t \leq \sigma_a^2 n_a / 4$,

$$\begin{aligned}
\Pr\left\{ n_a | \tau_{a;e}^2 - \mathbb{E}[\tau_{a;e}^2]| \geq t \right\} \leq& 2\exp\left( -\frac{1}{8c} \min\left\{ \frac{t^2}{\|\mathbf{P}_{n_a}\|_F^2 \sigma_a^2}, \frac{t}{\|\mathbf{P}_{n_a}\|} \right\} \right) \\
\leq& 2\exp\left( -\frac{1}{8c} \frac{t^2}{\|\mathbf{P}_{n_a}\|_F^2 \sigma_a^2} \right) \\
\leq& 2\exp\left( -\frac{t^2}{8c n_a \sigma_a^2} \right),
\end{aligned} \tag{13}$$

where $c$ is some constant, the 2nd step is from that $\frac{t^2}{\|\mathbf{P}_{n_a}\|_F^2 \sigma_a^2} < \frac{t}{\|\mathbf{P}_{n_a}\|}$ when $t \leq \sigma_a^2 n_a / 4$, due to $\|\mathbf{P}_{n_a}\|_F^2 / \|\mathbf{P}_{n_a}\| \geq n_a / 4$ for $n_a \geq 2$, and the last step is from $\|\mathbf{P}_{n_a}\|_F^2 < n_a$. Denoting

$\delta = 2 \exp\left(-\frac{t^2}{8cn_a\sigma_a^2}\right)$, we have that for $\delta \in \left[2\exp\left(-\frac{n_a\sigma_a^2}{32c}\right), 1\right)$,

$$\Pr\left\{|\tau_{a;e}^2 - \mathbb{E}[\tau_{a;e}^2]| \geq \sqrt{\frac{8c\sigma_a^2 \log(\delta/2)^{-1}}{n_a}}\right\} \leq \delta.$$

## A.5  Two lemmas

Motivated by the analysis on the regret bounds for Thompson Sampling in [15], [31] proved the following lemma which supplies an upper bound of the expected $n$-round regret under sampling distribution in general.

**Lemma 1** *Let the history of action $a$ after $n_a$ plays be a vector $H_{a,n_a}$ of length $n_a$. without loss of generality we assume that action 1 is optimal. Let $Y \sim dist(H_{a,n_a})$, where $dist(H_{a,n_a})$ is a sampling distribution depending on $H_{a,n_a}$. Define for $\eta \in \mathbb{R}$*

$$Q_{a,n_a}(\eta) = \Pr(Y \geq \eta | H_{a,n_a}).$$

*For $\eta \in \mathbb{R}$, the expected $n$-round regret can be bounded from above as*

$$R(n) \leq \sum_{a:\mu_a < \mu_*} \Delta_a(R_a^{(1)} + R_a^{(2)}),$$

*where*

$$R_a^{(1)} = \sum_{n_1=0}^{n-1} \mathbb{E}\left[\min\left\{\frac{1}{Q_{1,n_1}(\eta)} - 1, n\right\}\right] \text{ and } R_a^{(2)} = \sum_{n_a=0}^{n-1} \Pr[Q_{a,n_a}(\eta) > 1/n] + 1.$$

The following result is provide by [30].

**Lemma 2** *Let $X$ be a mean zero random vector in $\mathbb{R}^n$. If $X$ has the convex concentration property with constant $K$, then for any $n \times n$ matrix $A$ and every $t > 0$,*

$$Pr\left(|X^\top A X - E(X^\top A X)| \geq t\right) \leq 2\exp\left(-\frac{1}{CK^2}\min\left\{\frac{t^2}{\|A\|_F^2 \|Cov(X)\|}, \frac{t}{\|A\|}\right\}\right)$$

*for some universal constant $C$.*

## A.6  Derivation of $\mathbf{E}(\tau_a^2)$

By simple calculation,

$$\begin{aligned}\text{trace}(\mathbf{P}_{n_a}) &= \text{trace}\left(\mathbf{I}_n - 2\mathbf{Q}_{n_a} + \mathbf{Q}_{n_a}^\top \mathbf{Q}_{n_a}\right) \\ &= n_a - (1 + 1/2 + \cdots + 1/n_a).\end{aligned}$$

Noting $\mathrm{E}(\tau_a^2) = n_a^{-1}\sigma_a^2\text{trace}(\mathbf{P}_{n_a})$ and $\log(n_a + 1) < 1 + 1/2 + \cdots + 1/n_a \leq 1 + \log(n_a)$, we have that $\sigma_a^2\left[1 - \frac{\log(n_a)}{n_a}\right] \leq \mathrm{E}(\tau_a^2) < \sigma_a^2\left[1 - \frac{1+\log(1+n_a)}{n_a}\right]$. It follows that as $n_a \to \infty$,

$$\mathrm{E}(\tau_a^2) - \sigma_a^2 \to 0.$$

## A.7  Expected Regret Analysis

In this section, we derive an upper bound on the expected regret of the algorithms in Algorithm 1 showing that the optimal logarithmic regrets are achievable uniformly over time. Define $\Delta_a = \mu_* - \mu_a$, where, we recall that $\mu_a$ is the expected reward for action $a$ and $\mu_* = \max\{\mu_1, \cdots, \mu_K\}$.

Consider the algorithm SAU-Sampling. Thanks to the fact that $Y_a$ in Algorithm 1 is sampled from a Gaussian, we can derive an upper bound on the expected $n$-round regret of Algorithm 1 by applying the Gaussian tail bound.

**Theorem 1** *If SAU-Sampling is run on a $K$-armed bandit problem ($K \geq 2$) having arbitrary reward distribution $P_1, \cdots, P_K$ with support in $[0,1]$, then its expected regret after any number $n \geq \max\{\frac{24 \log n}{\min_a \Delta_a^2}, 4\}$ of plays is at most*

$$\sum_{a:\mu_a < \mu_*} \Delta_a \left( \frac{96 \log n}{\Delta_a^2} + 6 \right).$$

A proof of Theorem 1 is provided in Section A.8 below.

In SAU-UCB, $\tau_{a_n}^2$ is updated at every $n$, but to simplify the analysis we replace $\tau_{a_n}^2$ with a constant $\tau_a^{*2}$. This is justified thanks to Proposition 3 which says that $\tau_{a_n}^2$ is concentrated around its expectation, which tends to $\mu_a^2 + \sigma_a^2$ for growing $n_a$. With this simplification, we can directly reuse the regret analysis of UCB1, and obtain the following Theorem.

**Theorem 2** *If SAU-UCB with fixed $\tau_{a_n}^2 = \tau_a^{*2}$ is run on a $K$-armed bandit problem ($K \geq 2$) with arbitrary reward distributions $P_1, \ldots, P_K$ with support in $[0,1]$, then for any $\kappa \geq 2/\min\{\tau_a^*\}_{a=1}^K$, its expected regret after an arbitrary number $n$ of plays is at most*

$$\sum_{a:\mu_a < \mu_*} \Delta_a \left( \frac{4 \log n}{\Delta_a^2} + 1 + \frac{\pi^2}{3} \right).$$

Generalization of this regret analysis in the case where the constant $\tau_{a_n}^2$ assumption is invalid is left for future work.

## A.8 Proof of Theorem 1

**Proof** We prove Theorem 1 by applying the Gaussian tail bound and Lemma 1, which is attached in the Appendix. In what follows, without loss of generality we assume that action 1 refers to the optimal action, i.e., $\mu_1 = \mu_*$. Fix $n$, for simplifying notations, we drop the subindex of $n$ in quantities if without confusion. Let $r_{a,1}, r_{a,2}, \cdots, r_{a,n_a}$ be the random variables referring to the rewards yielded by action $a$ of successive $n_a$ plays. The notation "$a, j$" means that the number of plays of action $a$ is $j$. $Y_a$ is random variable drawn from the normal distribution $\mathcal{N}(\hat{\mu}_{a,n_a}, \tau_a^2/n_a)$, where

$$\hat{\mu}_{a,n_a} = \frac{1}{n_a} \sum_{j=1}^{n_a} r_{a,j} \quad \text{and} \quad \tau_a^2 = \frac{1}{n_a} \sum_{j=1}^{n_a} (r_{a,j} - \hat{\mu}_{a,j-1})^2.$$

Rewrite

$$Y_a = \hat{\mu}_{a,n_a} + \delta_{a,n_a}, \quad \text{where } \delta_{a,n_a} \sim \mathcal{N}(0, \tau_a^2/n_a). \tag{14}$$

We bound $\delta_{a,n_a}$ in probability. By the Gaussian tail bound (Fact 2), for $\alpha > 0$,

$$\Pr\left\{\delta_{a,n_a} \geq \alpha \mid \tau_a^2\right\} \leq \exp\left\{-\frac{\alpha^2 n_a}{2\tau_a^2}\right\} \leq \exp\left\{-n_a \alpha^2/2\right\}, \tag{15}$$

where the second inequality is from that $\tau_a^2 \leq 1$. Eqn. (15) follows that

$$\Pr\left\{\delta_{a,n_a} \geq \alpha\right\} \leq \exp\left\{-n_a \alpha^2/2\right\}. \tag{16}$$

Denote $c_{1,n} = \sqrt{n_1^{-1} \log n}$ and $c_{a,n} = \sqrt{n_a^{-1} \log n}$. Let

$$A_1 = \{\hat{\mu}_{1,n_1} > \mu_1 - c_{1,n}\}, \text{ and } A_a = \{\hat{\mu}_{a,n_a} < \mu_a + c_{a,n}\} \text{ for } a = 2, \cdots, K.$$

Denote $\bar{A}_1$ and $\bar{A}_a$ be the complements of $A_1$ and $A_a$ respectively. $\Pr\left\{\bar{A}_1\right\}$ and $\Pr\left\{\bar{A}_a\right\}$ are bounded from the Azuma-Hoeffding inequality (Fact 1):

$$\Pr\left\{\bar{A}_1\right\} = \Pr\{\hat{\mu}_{1,n_1} \leq \mu_1 - c_{1,n}\} \leq \exp(-2 \log n) = n^{-2}; \tag{17}$$

$$\Pr\left\{\bar{A}_a\right\} = \Pr\{\hat{\mu}_{a,n_a} \geq \mu_a + c_{a,n}\} \leq \exp(-2 \log n) = n^{-2}. \tag{18}$$

Define for $\eta \in \mathbb{R}$

$$Q_{a,n_a}(\eta) = \Pr(Y_a \geq \eta).$$

Lemma 1, which is proved by [31], shows that

$$R(n) \leq \sum_{a:\mu_a < \mu_*} \Delta_a (R_a^{(1)} + R_a^{(2)}),$$

where

$$R_a^{(1)} = \sum_{n_1=0}^{n-1} \mathbb{E}\left[\min\left\{\frac{1}{Q_{1,n_1}(\eta)} - 1, n\right\}\right] \text{ and } R_a^{(2)} = \sum_{n_a=0}^{n-1} \Pr[Q_{a,n_a}(\eta) > 1/n] + 1.$$

We set

$$\eta_a = \mu_a + \frac{\Delta_a}{2} = \mu_1 - \frac{\Delta_a}{2}. \tag{19}$$

Our proof is to bound the terms $R_a^{(1)}$ and $R_a^{(2)}$ by applying Lemma 1. Now in the first step we derive the upper bound on $R_a^{(1)}$. Denote $\bar{n}_a = \frac{24 \log n}{\Delta_a^2}$. Noting $Q_{1,0}(\eta_a) = 1$,

$$
\begin{aligned}
R_a^{(1)} &= \sum_{n_1=1}^{n-1} \mathbb{E}\left[\min\left\{\frac{1}{Q_{1,n_1}(\eta_a)} - 1, n\right\}\right] \\
&= \sum_{n_1=1}^{\lceil \bar{n}_a \rceil - 1} \mathbb{E}\left[\min\left\{\frac{1}{Q_{1,n_1}(\eta_a)} - 1, n\right\}\right] + \sum_{n_1=\lceil \bar{n}_a \rceil}^{n-1} \mathbb{E}\left[\min\left\{\frac{1}{Q_{1,n_1}(\eta_a)} - 1, n\right\}\right] \\
&=: R_a^{(1)L} + R_a^{(1)U},
\end{aligned} \tag{20}
$$

where $\lceil x \rceil$ is the smallest integer not less than $x$.

The law of total probability implies that, when $n_1 \geq \bar{n}_a$,

$$
\begin{aligned}
Q_{1,n_1}(\eta_a) =&\Pr\left\{Y_1 > \eta_a\right\} = 1 - \Pr\left\{Y_1 < \eta_a\right\} \\
=&1 - \Pr(A_1)\Pr\left\{Y_1 < \eta_a | A_1\right\} - \Pr(\bar{A}_1)\Pr\left\{Y_1 < \eta_a | \bar{A}_1\right\} \\
>&1 - \Pr\left\{\hat{\mu}_{1,n_1} + \delta_{1,n_1} < \mu_1 - \Delta_a/2 | A_1\right\} - \Pr(\bar{A}_1) \\
\geq&1 - \Pr\left\{\delta_{1,n_1} \leq -\Delta_a/2 + c_{1,n}\right\} - \Pr(\bar{A}_1) \\
>&1 - \Pr\left\{\delta_{1,n_1} \leq -\sqrt{2}c_{1,n}\right\} - \Pr(\bar{A}_1) \\
\geq&1 - n^{-1} - n^{-2},
\end{aligned} \tag{21}
$$

where the 1st inequality is from the facts that $\Pr(A_1) < 1$ and $\Pr\left\{Y_1 < \eta_a | \bar{A}_1\right\} < 1$, the 2nd inequality is from the definition of $A_1$, the 3rd inequality is from $\Delta_a/2 - c_{1,n} \geq \sqrt{2}c_{1,n}$ as $n_1 \geq \bar{n}_a$, and the last inequality is from Eqns. (16) and (17).

Eqn. (21) implies that for $n \geq 3$,

$$
\begin{aligned}
R_a^{(1)U} &= \sum_{n_1=\lceil \bar{n}_a \rceil}^{n-1} \mathbb{E}\left[\min\left\{\frac{1}{Q_{1,n_1}(\eta_a)} - 1, n\right\}\right] = \sum_{n_1=\lceil \bar{n}_a \rceil}^{n-1} \min\left\{\frac{1}{Q_{1,n_1}(\eta_a)} - 1, n\right\} \\
&< \sum_{n_1=\lceil \bar{n}_a \rceil}^{n-1} \frac{1}{1 - n^{-1} - n^{-2}} - 1 < 2 \sum_{n_1=\lceil \bar{n}_a \rceil}^{n-1} (n^{-1} + n^{-2}) < 4,
\end{aligned} \tag{22}
$$

where the 1st inequality is from that $\frac{1}{Q_{1,n_1}(\eta_a)} - 1 < \frac{1}{1-n^{-1}-n^{-2}} - 1 < n$, the 2nd inequality is from that $\frac{1}{1-n^{-1}-n^{-2}} - 1 < 2(n^{-1} + n^{-2})$ when $n \geq 3$.

Next we investigate the term $R_a^{(1)L}$ in Eqn. (20). For it, we now provide another lower bound of the $Q_{1,n_1}(\eta_a)$. Let

$$A_1^L = \{\hat{\mu}_{1,n_1} \geq \mu_1\}.$$

Denote $\bar{A}_1^L$ be the complements of $A_1^L$. We have that

$$\Pr\left\{A_1^L\right\} = 1/2; \ \ \Pr\left\{\bar{A}_1^L\right\} = 1/2 \tag{23}$$

Similarly as Eqn. (21), we have that

$$
\begin{aligned}
Q_{1,n_1}(\eta_a) =& \Pr\left\{Y_{1,n_1} > \eta_a\right\} \\
=& 1 - \Pr(A_1^L)\Pr\left\{Y_{1,n_1} < \eta_a | A_1^L\right\} - \Pr(\bar{A}_1^L)\Pr\left\{Y_{1,n_1} < \eta_a | \bar{A}_1^L\right\} \\
>& 1 - 1/2\Pr\left\{\hat{\mu}_{1,n_1} + c_{1,n}\delta_{1,n_1} < \mu_1 - \Delta_a/2 | A_1^L\right\} - \Pr(\bar{A}_1^L) \\
\geq& 1/2 - 1/2\Pr\left\{c_{1,n}\delta_{1,n_1} \leq -\Delta_a/2\right\} \\
\geq& \frac{1}{2}\left[1 - \exp\left(-\frac{n_1\Delta_a^2}{8\log n}\right)\right],
\end{aligned}
\tag{24}
$$

where the 1st inequality is from the facts that $\Pr\left\{Y_{1,n_1} < \eta_a | \bar{A}_1^L\right\} < 1$, and the 2nd inequality is from the definition of $A_1^L$ and Eqn. (23), and the last inequality is from Eqn. (16).

We have that (1) $\log(1 - \frac{2}{n+1}) \geq \frac{-3}{n+1}$ when $n \geq 4$; and (2) $-\frac{\Delta_a^2}{8\log n} \leq \frac{-3}{n+1}$ when $\frac{n}{\log n} \geq \frac{24}{\min_a \Delta_a^2}$. The two inequalities imply that when $n \geq \max\{\frac{24\log n}{\min_a \Delta_a^2}, 4\}$,

$$
1 - \exp\left(-\frac{\Delta_a^2}{8\log n}\right) \geq \frac{2}{n+1}.
$$

Thus, Eqn. (24) follows that

$$
\begin{aligned}
R_a^{(1)L} =& \sum_{n_1=1}^{\lceil \bar{n}_a \rceil - 1} \mathbb{E}\left[\min\left\{\frac{1}{Q_{1,n_1}(\eta_a)} - 1, n\right\}\right] \\
\leq& \sum_{n_1=1}^{\lceil \bar{n}_a \rceil - 1}\left[\frac{2}{1 - \exp\left(-\frac{n_1\Delta_a^2}{8\log n}\right)} - 1\right] \\
<& \sum_{n_1=1}^{\lceil \bar{n}_a \rceil - 1}\left[4\exp\left(-\frac{n_1\Delta_a^2}{8\log n}\right) - 1\right] \\
<& 3\bar{n}_a.
\end{aligned}
\tag{25}
$$

Therefore, inserting Eqns. (22) & (25) into Eqn. (20),

$$
R_a^{(1)} \leq 3\bar{n}_a + 4.
\tag{26}
$$

In the next step we derive the upper bound on $R_a^{(2)}$. When $n_a \geq \bar{n}_a$,

$$
\Delta_a/2 - c_{a,n} \geq \sqrt{2}c_{a,n}.
\tag{27}
$$

From the definition of event $A_a$, the law of total probability implies that

$$
\begin{aligned}
\Pr\left\{Q_{a,n_a}(\eta_a) > 1/n\right\} =& \Pr(A_a)\Pr\left\{Q_{a,n_a}(\eta_a) > 1/n | A_a\right\} + \Pr(\bar{A}_a)\Pr\left\{Q_{a,n_a}(\eta_a) > 1/n | \bar{A}_a\right\} \\
\leq& \Pr\left\{Q_{a,n_a}(\eta_a) > 1/n | A_a\right\} + \Pr(\bar{A}_a).
\end{aligned}
\tag{28}
$$

When $n_a \geq \bar{n}_a$, given event $A_a$,

$$
\begin{aligned}
Q_{a,n_a}(\eta_a) =& \Pr\left\{Y_a > \eta_a | A_a\right\} = \Pr\left\{\hat{\mu}_a + \delta_{a,n_a} > \Delta_a/2 + \mu_a | A_a\right\} \\
\leq& \Pr\left\{\delta_{a,n_a} \geq \Delta_a/2 - c_{a,n}\right\} \\
\leq& \Pr\left\{\delta_{a,n_a} \geq \sqrt{2}c_{a,n}\right\} \\
\leq& \exp\left\{-\log n\right\} = n^{-1}.
\end{aligned}
\tag{29}
$$

where the 1st inequality is from the definition of event $A_a$, the 2nd inequality is from Eqn. (27), the 3rd inequality is from Eqn. (16).

Eqn. (29) follow that when $n_a \geq \bar{n}_a$,

$$
\Pr\left\{Q_{a,n_a}(\eta_a) > 1/n | A_a\right\} = 0.
\tag{30}
$$

Inserting Eqn. (30) into Eqn. (28),

$$\Pr[Q_{a,n_a}(\eta_a) > 1/n] \leq \Pr(\bar{A}_a) \leq n^{-1}, \tag{31}$$

where the last step is from Eqn. (18).

We have that

$$
\begin{aligned}
R_a^{(2)} &= \sum_{n_a=0}^{n-1} \Pr[Q_{a,n_a}(\eta_a) > 1/n] + 1 \\
&\leq \bar{n}_a + \sum_{n_a=\lceil \bar{n}_a \rceil}^{n-1} \Pr[Q_{a,n_a}(\eta_a) > 1/n] + 1 \\
&< \bar{n}_a + 2,
\end{aligned}
\tag{32}
$$

where the 1st inequality is from direct calculation, the 2nd inequality is from Eqn. (31). Based on the bounds of $R_a^{(1)}$ in Eqn. (20) and $R_a^{(2)}$ in Eqn. (32), we have the statement

$$R(n) < \sum_{a=2}^{K} \Delta_a \left( \frac{96 \log n}{\Delta_a^2} + 6 \right).$$

∎

# Appendix B   Performance of linear contextual bandits in other settings

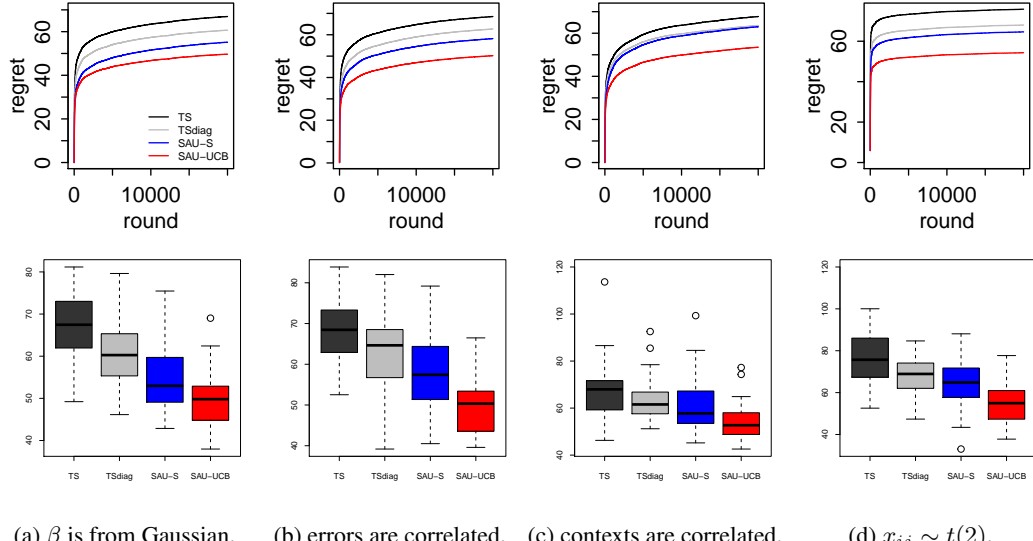

(a) $\beta$ is from Gaussian.     (b) errors are correlated.    (c) contexts are correlated.      (d) $x_{ij} \sim t(2)$.

Figure 3: Performance of other cases: the elements of $\beta$ are from $\mathcal{N}(0,1)$ then are normalized; the model errors are correlated; and the contexts are correlated; and the elements of $\mathbf{x}_i$ are from a heavy-tailed truncated $t$-distribution with $df = 2$.

## Appendix C   Wheel Bandit Problem

The *Wheel Bandit Problem* is a synthetic bandit designed by [18] where the effect of need for exploration can be systematically studied by varying a parameter $\delta \in [0, 1]$. The difficulty of the problem increases with $\delta$, since the problem is designed so that for $\delta$ close to 1 most contexts have the same optimal action, while only for a fraction $1 - \delta^2$ of contexts the optimal action is a different more rewarding action (see [18] for more details).

Table 2: Cumulative regret incurred on the Wheel Bandit Problem by the 4 best algorithms identified by [18] compared to our SAU-based algorithms (in bold) with increasing values of $\delta$. We report the mean and standard error of the mean over 50 trials. The best cumulative regret value for each task is marked in bold.

| $\delta$ | 0.50 | 0.70 | 0.90 | 0.95 | 0.99 |
|---|---|---|---|---|---|
| Linear | 60.62 ± 3.55 | 83.56 ± 5.21 | 84.11 ± 4.72 | 86.71 ± 4.08 | 93.61 ± 2.74 |
| LinearGreedy | 88.95 ± 0.19 | 124.38 ± 0.46 | 122.18 ± 1.05 | 119.85 ± 1.55 | 99.59 ± 2.61 |
| NeuralGreedy | 35.00 ± 0.90 | 56.65 ± 1.50 | 72.04 ± 1.56 | 83.97 ± 2.53 | 91.18 ± 1.91 |
| NeuralLinear | 18.71 ± 2.05 | 26.63 ± 1.21 | 45.47 ± 2.91 | 65.44 ± 3.23 | **86.03 ± 3.03** |
| **SAU-Linear-S** | 34.74 ± 1.63 | 42.69 ± 2.27 | 54.88 ± 3.01 | 71.37 ± 3.57 | **87.38 ± 2.85** |
| **SAU-Linear-UCB** | 42.23 ± 1.55 | 60.08 ± 2.12 | 78.51 ± 4.93 | 92.42 ± 4.67 | 96.79 ± 2.50 |
| **SAU-Neural-S** | **12.49 ± 4.11** | **13.72 ± 0.85** | **36.54 ± 3.03** | **63.30 ± 3.53** | 106.89 ± 14.17 |
| **SAU-Neural-UCB** | 26.29 ± 2.32 | 51.92 ± 2.03 | 73.87 ± 4.58 | 75.95 ± 2.24 | 89.99 ± 2.65 |
| Uniform | 100.00 ± 28.43 | 100.00 ± 0.46 | 100.00 ± 0.86 | 100.00 ± 1.41 | 100.00 ± 2.75 |

Table 3: Same results as the Table 2 above, but not normalized with respect to the uniform exploration strategy

| $\delta$ | 0.50 | 0.70 | 0.90 | 0.95 | 0.99 |
|---|---|---|---|---|---|
| Linear | 49742.3 ± 2911.4 | 33109.1 ± 2063.5 | 12816.6 ± 719.3 | 6954.2 ± 327.0 | 1864.8 ± 54.6 |
| LinearGreedy | 72982.5 ± 158.6 | 49287.7 ± 183.8 | 18617.9 ± 160.2 | 9611.9 ± 124.3 | 1983.9 ± 51.9 |
| NeuralGreedy | 28722.7 ± 739.4 | 22448.0 ± 592.6 | 10977.4 ± 237.2 | 6733.9 ± 203.1 | 1816.5 ± 38.0 |
| NeuralLinear | 15350.4 ± 1685.2 | 10550.6 ± 479.5 | 6929.1 ± 443.4 | 5248.1 ± 259.2 | **1713.9 ± 60.4** |
| **SAU-Linear-S** | 28506.8 ± 1336.8 | 16914.3 ± 899.3 | 8362.1 ± 459.0 | 5723.6 ± 286.2 | **1740.8 ± 56.9** |
| **SAU-Linear-UCB** | 34648.4 ± 1275.8 | 23806.7 ± 839.6 | 11962.9 ± 751.8 | 7411.6 ± 374.3 | 1928.2 ± 49.8 |
| **SAU-Neural-S** | **10247.5 ± 3369.4** | **5437.5 ± 335.5** | **5568.3 ± 461.1** | **5076.8 ± 283.1** | 2129.3 ± 282.2 |
| **SAU-Neural-UCB** | 21572.5 ± 1903.9 | 20571.7 ± 803.7 | 11256.4 ± 697.7 | 6091.3 ± 179.7 | 1792.8 ± 52.9 |
| Uniform | 82053.3 ± 23324.5 | 39625.3 ± 182.9 | 15238.1 ± 130.8 | 8019.8 ± 113.2 | 1992.1 ± 54.7 |

## Appendix D   Real-world Datasets

Here below we list all the dataset used to reproduce the deep bandit experiments in [18]. Most datasets are from he UCI Machine Learning Repository [32] and were manually inspected to verify that they do not contain personally identifiable information or offensive content.

**Mushroom.**   The Mushroom Dataset [33] has $N = 8124$ samples with $d = 117$ features (one-hots from 22 categorical attributes), divided in 2 classes ('poisonous' and 'edible'). As in [34], the bandit problem has $k = 2$ actions: 'eat' mushroom or 'pass'. If sample is in class 'edible' action 'eat' gets reward $r = +5$. If sample is in class 'poisonous' action 'eat' gets reward $r = +5$ with probability 1/2 and reward $r = -35$ otherwise. Action 'pass' gets reward of 0 in all cases. We use a number of steps of $T = 50000$.

**Statlog.**   The Shuttle Statlog Dataset [32] has $N = 43500$ samples with $d = 9$ features, divided in 7 classes. In the bandit problem the agent has to select the right class ($k = 7$ actions), in which case it gets reward $r = 1$, and $r = 0$ otherwise. The number of steps is $T = 43500$. Note: one action is the optimal one in 80% of cases, meaning that a successful algorithm has to avoid committing to this action and explore instead.

**Covertype.**   The Covertype Dataset [32] has $N = 581012$ samples with $d = 54$, divided in 7 classes. In the bandit problem the agent has to select the right class ($k = 7$ actions), in which case it gets reward $r = 1$, and $r = 0$ otherwise. The number of steps is $T = 50000$.

**Financial.**   The Financial Dataset was created as in [18] by pulling stock prices of 21 publicly traded companies in NYSE and Nasdaq between 2004 and 2018. The dataset has $N = 3713$ samples with $d = 21$ features. The arms are synthetically created to be a linear combination of the features, representing $k = 8$ different portfolios. The number of steps is $T = 3713$. Note: this is a very the small horizon, meaning that algorithms may over-explore at the beginning with no time to amortize the initial regret.

**Jester.**   The Jester Dataset [35] is preprocessed following [18] to have $N = 19181$ samples with $d = 32$ features associated to 8 continuous outputs. The agent selects one of the $k = 8$ outputs and obtains the reward corresponding to selected output. The number of steps is $T = 19181$.

**Adult.**   The training partition of Adult Dataset [32] has $N = 30162$ samples (after dropping rows with NA values). As in [18] we turn the dataset into a contextual bandit by considering the $k = 14$ occupations as feasible actions to be selected based on the remaining $d = 92$ features (after encoding categorical variables into one-hots). The agent gets a reward of $r = 1$ for making the right prediction, and $r = 0$ otherwise. The number of steps is $T = 30162$.

**Census.**   The US Census (1990) Dataset [32] has $N = 2458285$ samples with $d = 387$ features (after encoding categorical features to one-hots). The goal in the bandit tasks is to predict the occupation feature among $k = 9$ classes. The agent obtains reward $r = 1$ for making the right prediction, and $r = 0$ otherwise. The number of steps is $T = 25000$.

# Appendix E   Additional algorithms and results

---

**Algorithm 5** $\epsilon$-greedy exploration for Deep Contextual Bandits (NeuralGreedy)

---

1: **function** PREDICT($\mathbf{x}_n$)
2:     **Return** predicted values $\{\hat{\mu}_{n,a}\}_{a\in\mathbb{K}} = \mu(\mathbf{x}_n, \hat{\boldsymbol{\theta}}_n)$;

3: **function** ACTION($\{\hat{\mu}_{n,a}\}_{a\in\mathbb{K}}$)
4:     Compute $a^* = \arg\max_a(\{\tilde{\mu}_{n,a}\}_{a\in\mathbb{K}})$;
5:     **Return** $a_n = \begin{cases} a^*, \text{ with probability } 1-\epsilon \\ a \in \mathbb{K} \text{ uniformly at random, otherwise;} \end{cases}$     (**Exploration**)

6: **function** UPDATE($r_n, a_n, \mathbf{x}_n$)
7:     Compute loss $l_n = (r_n - \hat{\mu}_{n,a_n})^2$;
8:     Update model parameters $\hat{\boldsymbol{\theta}}_{n+1}$ using SGD with gradients $\frac{\partial l_n}{\partial \hat{\boldsymbol{\theta}}_n}$ (or mini-batch version);
9:     Update exploration parameter: decrease $\epsilon$ according to annealing schedule;

---

Table 4: Cumulative regret incurred on the contextual bandits in [18] by the 4 best algorithms that they identified and described in Appendix D compared to our SAU-based algorithms (in bold). This table shows the same results as Table 1 without normalizing to the Uniform algorithm. We report the mean and standard error of the mean over 50 trials. The best cumulative regret value for each task is marked in bold.

|  | Mushroom | Statlog | Covertype | Financial | Jester | Adult | Census |
|---|---|---|---|---|---|---|---|
| LinearPosterior | 7373.7 ± 376.1 | 3837.4 ± 70.0 | 15807.6 ± 28.2 | 509.4 ± 7.3 | 61674.5 ± 383.8 | 31861.4 ± 26.7 | 10380.0 ± 17.6 |
| LinearGreedy | 11332.3 ± 862.1 | 40725.9 ± 354.8 | 22933.0 ± 858.6 | 731.9 ± 48.4 | 60427.6 ± 433.7 | 36409.3 ± 99.9 | 14525.9 ± 348.7 |
| NeuralLinear | 6503.8 ± 200.8 | 469.7 ± 9.6 | 12509.9 ± 28.1 | 480.7 ± 8.6 | 66948.0 ± 411.1 | 32874.2 ± 30.3 | 9113.7 ± 19.9 |
| NeuralGreedy | 65153.3 ± 771.9 | 14974.8 ± 173.3 | 38085.6 ± 53.5 | 4060.0 ± 64.7 | 89748.9 ± 740.6 | 41559.7 ± 11.3 | 19075.3 ± 57.9 |
| **Linear-SAU-S** | 11188.6 ± 954.0 | 3892.8 ± 97.2 | 15554.1 ± 49.5 | 411.9 ± 24.8 | 60304.6 ± 413.2 | 31370.9 ± 55.6 | 8812.4 ± 25.5 |
| **Linear-SAU-UCB** | 7563.5 ± 375.3 | 3736.8 ± 80.5 | 15761.4 ± 74.7 | 307.7 ± 17.7 | 62078.3 ± 381.8 | **31337.0 ± 29.2** | 8885.6 ± 20.7 |
| **Neural-SAU-S** | **5381.1 ± 131.0** | 232.6 ± 5.2 | **11771.5 ± 26.9** | 264.7 ± 5.3 | **58795.1 ± 548.8** | 32765.8 ± 30.7 | 8614.5 ± 18.2 |
| **Neural-SAU-UCB** | 5665.7 ± 143.5 | **225.0 ± 5.5** | 11958.2 ± 24.5 | **248.8 ± 7.1** | 59997.0 ± 586.6 | 32793.4 ± 27.3 | **8565.7 ± 19.6** |
| Uniform | 244431.4 ± 583.4 | 37275.8 ± 9.9 | 42864.5 ± 9.6 | 4729.0 ± 69.4 | 96353.3 ± 923.2 | 41994.5 ± 7.1 | 22226.4 ± 8.0 |

Table 5: Running times for the simulations in Tables 1 and 4 run on Intel(R) Xeon(R) Gold 6248 CPU @ 2.50GHz. Results are averaged over 3 trials and the corresponding standard errors are also shown. Simulation time of SAU algorithms is consistently close the corresponding epsilon-greedy algorithm, confirming its computational efficiency.

|  | Mushroom | Statlog | Covertype | Financial | Jester | Adult | Census |
|---|---|---|---|---|---|---|---|
| LinearPosterior | 1203.2 s ± 4.9 | 384.0 s ± 1.9 | 900.5 s ± 37.5 | 10.8 s ± 0.2 | 115.0 s ± 3.1 | 1554.1 s ± 8.7 | 7273.3 s ± 31.3 |
| LinearGreedy | 542.6 s ± 6.7 | 334.0 s ± 7.9 | 406.4 s ± 0.8 | 3.0 s ± 0.0 | 56.4 s ± 1.0 | 359.5 s ± 3.8 | 141.9 s ± 1.1 |
| NeuralGreedy | 112.5 s ± 4.0 | 70.7 s ± 1.2 | 85.8 s ± 1.4 | 6.0 s ± 0.1 | 28.5 s ± 0.1 | 87.3 s ± 2.5 | 78.4 s ± 0.4 |
| NeuralLinear | 790.6 s ± 23.0 | 1032.4 s ± 44.8 | 1282.5 s ± 13.8 | 80.0 s ± 0.9 | 452.5 s ± 13.8 | 1997.1 s ± 6.8 | 784.2 s ± 5.7 |
| **Linear-SAU-S** | 463.6 s ± 1.1 | 332.3 s ± 9.0 | 479.7 s ± 70.3 | 3.2 s ± 0.0 | 55.6 s ± 0.5 | 335.4 s ± 24.9 | 150.6 s ± 3.5 |
| **Linear-SAU-UCB** | 460.2 s ± 2.2 | 338.8 s ± 8.0 | 398.1 s ± 1.5 | 3.2 s ± 0.0 | 56.6 s ± 1.3 | 308.3 s ± 0.9 | 144.7 s ± 0.6 |
| **Neural-SAU-S** | 116.4 s ± 3.9 | 77.5 s ± 2.2 | 99.2 s ± 3.4 | 6.3 s ± 0.6 | 29.7 s ± 0.3 | 86.0 s ± 3.0 | 81.8 s ± 0.4 |
| **Neural-SAU-UCB** | 115.6 s ± 4.2 | 81.0 s ± 2.0 | 97.5 s ± 1.4 | 6.1 s ± 0.4 | 29.3 s ± 0.1 | 85.6 s ± 3.3 | 82.1 s ± 0.1 |
| Uniform | 31.9 s ± 0.7 | 13.5 s ± 0.0 | 20.8 s ± 0.1 | 0.7 s ± 0.0 | 3.4 s ± 0.0 | 14.5 s ± 0.0 | 34.4 s ± 0.2 |

**Hyperparameters for deep contextual bandits**

Here we list the settings of the main hyperparameter of our simulations. For more details on how to reproduce our results, please refer to our released code.

- All neural network models are an MLP with 2 hidden layers of size 100 with ReLU activations.

- Training is generally done with Adam SGD [36] with $\beta_1 = 0.9$, $\beta_2 = 0.999$, and learning rate 0.003.

- The default model updating consisted in training for $t_s = 10$ mini-batches of size 64 every $t_f = 20$ steps.

- The hyperparameters for the competing Bayesian bandit algorithms are based on those mentioned in [18].

- For *Linear-SAU-Sampling* and *Linear-SAU-UCB* the parameter `lambda_prior` (the initial variance of the inverse-covariance matrix) is set to 20.0 for all datasets, apart from Financial (which is rather small) for which it is set to 0.25.

## Appendix F  Paper Checklist