# OpenReview forum: "Deep Bandits Show-Off: Simple and Efficient Exploration with Deep Networks"
_NeurIPS.cc/2021/Conference — NeurIPS 2021 Poster_

### Official Review · Reviewer_opYp · 2021-06-30

**Rating:** 6
**Confidence:** 4

**Summary:**

### Summary

(+) Simplicity!

(+) Formal justification and guarantees of the approach in two popular bandit settings

(+) The experiments on the bandit benchmark from [18] are convincing. I think it is meaningful to first consider and understand the bandit setting before the RL setting.

(+/-) The paper is overall well written and clear, except for some technical sections where I found the exposition confusing (see below)

(-) In my opinion, an important weakness appears to be the clear positioning of the paper in the light of previous research (see details below). The approach---which seems very similar to maximum likelihood + using the residual variance to drive exploration---should be better connected to previous work and the claimed contributions revised accordingly (perhaps only the evaluation and theoretical guarantees are novel?)


**Limitations And Societal Impact:**

Adequately addressed.

**Main Review:**

### Detailed comments

* The paper could explain more why it focuses on the instantaneous regret as opposed to the cumulative regret.

* I am a bit confused by the derivations in Sec. 3.1.: (a) Are the {r_n}\_{n in T_a} assumed to be independently distributed?  (b) For the expression of sigma^2_{n, a}, is the expectation with respect to distribution of rewards conditional on a_n=a? (c) How does the sequence of estimators {hat{theta}_{n,a}}\_{n in T_a} relate to theta_a, in particular to ensure that tau^2_a is an unbiased estimator?

* Is tau^2_a meant to capture the aleatoric uncertainty or the epistemic uncertainty? It seems to me that this is the former since its definition corresponds to the residual variance.

* I am under the impression that the proposed scheme corresponds to using the maximum-likelihood estimators (MLE)  (for a Gaussian likelihood  r | x, a ~ N(mu(x, theta_a), sigma^2_a)) of both the mean and the variance (we could even further simplify the reasoning by assuming that we store all the contexts so that the sample mean and variance could be evaluated at a single estimator hat{theta}_a per action a).
Using such a residual variance (i.e., more related to the aleatoric uncertainty) to drive exploration has been already proposed, e.g., [A] for Bayesian optimization (see their Sec. 4.1 that corresponds to SAU-Sampling) and [B] (when they reduce to a single ensemble member; see Fig. 1, second plot from the left)---Those two references are recent, there must be earlier ones.
I would therefore suggest to better connect the proposed formalism with previous works and MLE, and clearly state what the novel contributions are (e.g., the theoretical analysis?)

* In both Proposition 1 and 2, it would be helpful to specify with respect to which random variables the expectations are taken.

* “...meaning that contexts should be ‘homogeneous’ to a certain extent.”: If we were to assume a random design setting with x_n distributed according to some Gaussian distribution, what would be the probability of the assumption in Proposition 4 to hold?

* In Sec. 4.2., it seems that the proposed estimator (line 172) requires for all actions a, n_a > p in order to guarantee the existence of sum_j x_j x_j^T; is that the case?


* In Sec. 4.3, why aren't UCB and UCB-diag (based on the linear model, like TS) part of the  comparison?

* I would like to better understand what the effect is of computing the sample mean and variance for different estimators at each time step. An ablation could be run in which the triplets (r, a, x) are all recorded so that the sample mean and variance could be computed at a single (i.e., the latest) estimator.

* (I only skimmed through the proofs in the appendix and they look technically sound)


### Minor

* “...of the the action…” → remove “the”

* If space is not a concern, “ACT” could be replaced by the more readable “ACTION”

### References

[A] Salinas et al., A Quantile-based Approach for Hyperparameter Transfer Learning (https://arxiv.org/pdf/1909.13595.pdf)

[B] Lakshminarayanan et al., Simple and Scalable Predictive Uncertainty Estimation using Deep Ensembles (https://papers.nips.cc/paper/2017/file/9ef2ed4b7fd2c810847ffa5fa85bce38-Paper.pdf)


### UPDATE POST REBUTTAL

I have gone through the reviews and the replies of the authors, whom I would like to thank.

Most of my concerns/questions have been clarified, e.g.,
* The positioning with respect to previous research and maximum likelihood estimators is clearer. And I assume the authors would add the related discussion in their final version of the paper.
* Absence of UCB in the comparison of Sec. 4.3.
* Derivations in Sec. 3.1.

While those clarifications were leading me to increase my score further, the discussion with the other reviewers made me realise that I had overlooked the important missing baselines (e.g., NeuralUCB (Zhou et al. 2020)/NeuralTS (Zhang et al. 2021)) and the partial theoretical analysis. The combined (i) partial theoretical analysis together with (ii) the partial experimental comparisons are leading me to keep my score. The only weakness (i) would have been acceptable provided a stronger experimental comparison.


**Time Spent Reviewing:**

3.5

---

> ### Author Response · Authors · 2021-08-10
> **Response to Reviewer opYp**
>
> We thank the reviewer for the time spent examining our work, and for expressing positive comments on the simplicity of our method, the clarity of the exposition and for appreciating the setting that we chose for experimentally validating our algorithms.
>     Here below we address the comments of the reviewer and do our best to clarify any outstanding questions.
>
> * **Positioning of the paper in light of previous research like reference [A]:**
>
>     Thank you for pointing out where we've been unclear in highlighting the novel aspects of our contribution, in particular in relation to reference [A] mentioned by the reviewer.
>     Despite some superficial similarities, we maintain that our algorithm is fundamentally different from what is proposed in [A]. Although our exploration algorithms also has two parts, the prediction (which looks similar to mu(x) in [A]) and the bonus SAU term (which looks similar to sigma(x) in [A]), what these terms specifically implement are clearly distinct.
>     First, our prediction is not restricted to MLE, but can be a prediction from any arbitrary estimation method, such as one provided for instance by a Deep Bandit. In fact, we don't require any probabilistic assumption (in term for instance of likelihood) on the prediction provided by the model, which is why in the text we emphasized the "frequentist" character of SAU.
>     Second, our SAU quantity $\tau^2$ is simply the per-step squared prediction error, i.e., the average cumulative squared prediction error, not a direct estimate of the variance of the different arms. In fact, our $\tau^2$ does not rely on the traditional variance estimation used in the UCB literature, but is instead simply computed from the prediction. This makes for a very straightforward implementation that can be naturally applied to settings like deep networks.
>     In fact, SAU can be implemented as a drop-in module that can be added on top of any neural network model to endow it with the capability of explore, without needing to modify the architecture, the training procedure, and without adding any substantial computation overhead.
>     We will strive to further clarify this point and relation to existing work in the new version of the paper.
>
> * **Focus on the instantaneous regret as opposed to the cumulative regret:**
>
>     For showing the rationality of tau_a^2, we focus on the instantaneous tau value in order to show that it can efficient measure the instantaneous uncertainty.
>     For the cumulative regret, in Appendix A.7 we derive an upper bound of the cumulative regret on the expected cumulative regrets of SAU-UCB and SAU-Sampling in multi-armed bandits proving that the optimal logarithmic regrets are achievable uniformly over time.
>
> * **Derivations in Sec. 3.1:**
>
>     (a) Yes, the \{r_n\}\_\{n in T_a\} are assumed to be independently distributed.
>
>     (b) Yes, thanks for pointing out this potential source of confusion in the notation. For avoiding it, we may remove the "for a_n=a" in Line 100. We have defined the set of time-steps when action a was chosen. The quantity sigma^2_{n, a} is calculated based on this set.
>
>     c) For introducing tau_a^2 in general, we just assume that there is a sequence of estimators of theta_a. In specific problems, this sequence of estimators should be good in some sense. For example, in Sections 4.1 and 4.2, these estimators are unbiased.
>
> * Yes, tau^2_a tries to capture the aleatoric uncertainty due to the probabilistic nature of random rewards.
>
> * In Proposition 1 and 2, the expectations are taken with respect to the reward generating process (reward noise). Thank you for this suggestion. In the revised paper, we will clarify this point.
>
> * If we were to assume a random design setting with x_n distributed according to some Gaussian distribution, h_j would indeed be homogenous, i.e, h_j is close to p/n_a. In other words, for a constant p, the assumption in Proposition 4 holds.
>
> *  Yes, in Sec. 4.2 there is a default assumption that sum_j x_j x_j^T is invertible. But this assumption is with no loss of generality, since it can be guaranteed by changing sum_j x_j x_j^T into c*I+sum_j x_j x_j^T, where I is the identity matrix and c is a small positive constant. In the revised version, we will clarify this point.
>
> * In Sec. 4.3, we didn't report results on UCB and UCB-diag after noticing in preliminary experiments that UCB tends to be substantially worse than TS for contextual bandits. Since this observation turned out to be almost trivial in light of published literature we decided to completely omit the results on UCB. In the revised version, we will clarify this for completeness.
>
> * Our SAU quantity $\tau^2$ is updated just by the per-step squared prediction error. In other words, we just use the residuals that have been obtained at each step, rather than re-evaluating the prediction or residuals by the latest estimates (as suggested by the reviewer).
>     There are several reasons for which we opted for not re-evaluating the prediction residuals.
>
>     (1) This makes the implementation of our SAU quantity simpler and computation efficient, while recording (r, a, x) triplets and re-estimating residuals will cost additional computation and memory.
>
>     (2) From the viewpoint of statistical efficiency, as the training samples increases, the relative negative impact of potentially inaccurate early predictions and residuals will decrease, such that to re-evaluating them might actually minimally impact the quality of the uncertainty measure. From a theoretical standpoint, Proposition 3 indeed supports this intuition by formally proving that  $\tau^2$ is concentrated around its expectation, even if residuals are not re-computed.
>
>     (3) More importantly, our aim in this paper is to perform exploration. From the viewpoint of exploration, the impact of potentially inaccurate estimates in the initial steps may actually be beneficial, since they introduce noise that contributes to driving robust exploration. We think that the good performance of SAU in the empirical evaluations does indeed support this intuition.

---

### Official Review · Reviewer_bpP1 · 2021-07-03

**Rating:** 5
**Confidence:** 4

**Summary:**

This paper proposes a simple and efficient uncertainty measure Sample Average Uncertainty (SAU) for contextual bandits. The paper provides theoretical analysis to show that the uncertainty measure estimated by SAU asymptotically matches that estimated by Thompson Sampling. The paper also combines SAU with neural networks to design a deep contextual bandit algorithm which achieves good empirical performance. The presented experimental results on real-world datasets demonstrate that SAU-based exploration outperforms current state-of-the-art deep Bayesian bandit methods with low computation cost.

**Limitations And Societal Impact:**

Yes

**Main Review:**

Strengths:
1.	This paper conducts experiments on extensive real-world datasets to show the performance superiority of the proposed SAU-based exploration over existing deep Bayesian bandit methods.
2.	The proposed SAU based algorithms are simple and easy to implement. The computation cost is largely reduced compared to existing deep Bayesian bandit methods.

Weakness:
1.	Regret performance is one of the most important and canonical metrics in the bandit literature, but this paper only presents the regret bounds for the proposed SAU-UCB and SAU-Sampling algorithms for the preliminary K-armed bandit setting (in Appendix). The regret results of SAU-UCB/SAU-Sampling and SAU-Neural-UCB/SAU-Sampling for the contextual bandit problem are not provided, so readers cannot directly compare the theoretical performance of the proposed SAU-based algorithms with existing ones. For this reason, I give a ‘below the acceptance threshold’ score. If the authors could provide sufficient theoretical analysis of SAU-UCB/SAU-Sampling and SAU-Neural-UCB/SAU-Sampling for the (deep) contextual bandit problem, I am ready to raise my score.
2.	Comparisons with some important related works on theoretical and empirical results are missing. For example, the classic algorithm LinUCB (Abbasi-Yadkori et al. 2011) for contextual linear bandits, and the recent state-of-the-art algorithms NeuralUCB (Zhou et al. 2019) and NeuralTS (Zhang et al. 2020) for deep contextual bandits. This paper needs to include comparisons on regret bounds and empirical evaluations with these important related works, and give a sufficient discussion on why the SAU-based exploration can attain better/state-of-the-art performance on online regrets and computation costs.

References:
1.	Yasin Abbasi-Yadkori, Dávid Pál, and Csaba Szepesvári. Improved algorithms for linear stochastic bandits. NIPS, 2011.
2.	Dongruo Zhou, Lihong Li, and Quanquan Gu. Neural contextual bandits with UCB-based exploration. ICML, 2020.
3.	Weitong Zhang, Dongruo Zhou, Lihong Li, and Quanquan Gu. Neural thompson sampling. ICLR, 2021.


**After Rebuttal**

I have read the rebuttal of the authors.

Overall, I think that while the proposed SAU measure is simple and interesting, given that the theoretical analysis is not completed, this paper still needs further improvements and is not sufficient for acceptance in terms of its current version. So I keep my score ‘5’.

For the bandit literature, the regret is a critical performance metrics. I suggest the authors to complete the regret analysis for contextual bandits to demonstrate the superiority of the proposed SAU measures on regret performance.

As for the experiments, comparisons with recent deep algorithms are helpful for readers to understand the empirical performance of SAU-based deep algorithms. It would be better if the authors include these experimental results and comparisons in their revision.



**Time Spent Reviewing:**

4

---

> ### Author Response · Authors · 2021-08-10
> **Response to Reviewer bpP1**
>
> We want to thank the reviewer for the time spent reviewing our work and the comments provided that we can hopefully use to improve the revised version of the paper.
>     Here below we address the weaknesses pointed out by the reviewer.
>
> * The reviewer is correct that we do not provide a full regret analysis in the contextual bandits setting, although we give one for the multi-armed bandit case. This is a limitation of our theoretical results, that is due to the difficulty of characterizing the randomness of tau_a^2.
>     However, we would like to point out that Proposition 4 theoretically shows that tau_a^2 is a conservative measure of the uncertainty in terms of MSE in the linear contextual bandits. Despite not being a full-fledged regret analysis, we believe that this provides an important theoretical hint as to why SAU works in contextual bandits, and could be the basis for future theoretical work establishing a regret analysis in the contextual bandit setting. We would like to think as a regret analysis for the contextual bandit case as outside of the scope of our initial work, which is instead focused on proposing a new measure of uncertainty, and proving its practical scalability in deep neural network implementations (along with, nonetheless, theoretical regret guarantees in the multi-armed bandit case).
>
> * In preliminary experiments we noticed that UCB tended to work substantially worse than TS for contextual linear bandits. We then verified that this is a well-known fact amply reported in the previous literature, and therefore chose not to report experiments on LinUCB which could then be perceived as a well-known weak baseline. As said, this is also in line with the previous literature. In the revised version we will clarify this point.
>
> * We thank the reviewer for pointing out these recent references, NeuralUCB (Zhou et al. 2019) and NeuralTS (Zhang et al. 2020) for deep contextual bandits.
>     We will make sure to include citations in the revised paper, since they are somehow relevant, along with the following discussion.
>     The main distinction between NeuralUCB and SAU-UCB is that the two methods use fundamentally different measures of uncertainty.
>     As an extension of LinUCB, NeuralUCB designs an exploration algorithm by approximating the confidence interval of the prediction given by a neural network model. In contrast, SAU does not measure confidence interval, but drives exploration simply by measuring the sample average uncertainty, and plugging this quantity in either a UCB-like or a TS-like exploration strategy.
>     Compared with both NeuralUCB and NeuralTS, the biggest advantage of our method is its computational efficiency in implementing exploration.
>     In particular, both NeuralUCB and NeuralTS need to compute and update a "covariance matrix" which in practice is calculated in terms of the cumulative outer product of neural network gradients. The resulting memory, computational and implementation complexity of this procedure is remarkably high compared to SAU, specifically because of the need of building at each step the outer product between gradients of the neural network, which is a quantity that scales quadratically with the number of the neural network parameters and is therefore prohibitive for even moderately large models. There were some clever proposal for approximations which however still need a costly matrix inversion at each step.
>     We think that this type of algorithms additionally emphasizes the efficiency of our algorithm, and its minimal computational and memory requirements that allows it to scale up to even huge neural network models with a running time that is comparable to the simplest exploration strategies like epsilon greedy exploration, as we show in Table 3 in Appendix D.
>     Another important point that is worth pointing out is that the regret analyses of NeuralUCB and NeuralTS are only valid within the "lazy-training" NTK regime. Outside of that regime, which is arguably where the use of deep learning is most interesting and makes most sense, their regret guarantees are no longer valid, while we on the other hand still have the theoretical guarantee that SAU is a conservative measure of uncertainty in terms of MSE  (beside the regret bound guarantee in the multi-armed bandit case).
>     In the revised version, we will be happy to discuss these points and comparisons, and the main conclusion that NeuralUCB and NeuralTS are as of now limited in their scalability compared to SAU, and only offer stronger theoretical guarantees in the NTK regime.

---

> > ### Comment · Reviewer_bpP1 · 2021-08-15
> > **Response to Authors**
> >
> > Thank you for your reply.
> >
> > 1.	(Theoretical Analysis) What does the conclusion “tau_a^2 is a conservative measure” imply with respect to the regret performance of the algorithms? Do you mean that you can prove tau_a^2 is a conservative measure, but cannot (provably) determine that the algorithms using SAU for contextual bandits have lower regrets than those using UCB?
> > 2.	(Experiments) I knew that the focuses of NeuralUCB/NeuralTS and your SAU-Neural-UCB/SAU-Neural-Sampling are different and NeuralUCB/NeuralTS have big limitations on computation and memory complexity. I meant that considering that SAU-Neural-UCB/SAU-Neural-Sampling do not have theoretical analysis, you could compare them to NeuralUCB/NeuralTS in experiments to demonstrate the empirical performance, since empirical performance is important for deep algorithms. Is there any reason in implementation that you could not compare them?

---

> > > ### Author Response · Authors · 2021-08-17
> > > **Reply to additional questions**
> > >
> > > 1. Yes, that's right. We can prove that SAU is conservative, but that doesn't directly translate in a regret bound in the contextual bandit case. It might however serve as the basis for such a theoretical result or other theoretical results, since controlling SAU in terms of MSE ties uncertainty and exploration to the performance of the value function.
> > > 2. Following up on the reviewer's suggestion, we started replicating the main results of the NeuralTS and NeuralUCB paper. We already have the preliminary results that on Mushroom (in the way it was used in the NeuralTS paper, which is markedly different than the bandit version in Riquelme et al.) and Shuttle datasets, Neural-SAU-Sampling displays a performance that on average is more than 50% better than NeuralTS and NeuralUCB in terms of final cumulative regret. We will be happy to discuss these results more in detail in the revised version of the paper, if the reviewers deemed that useful for acceptance.

---

> > > > ### Comment · Reviewer_bpP1 · 2021-08-19
> > > > **Response to the Authors**
> > > >
> > > > Thank you for your replies and the supplementary experiments.
> > > >
> > > > Overall, I think that while the proposed SAU measure is simple and interesting, given that the theoretical analysis is not completed, this paper still needs further improvements and is not sufficient for acceptance in terms of its current version. So I keep my score ‘5’.
> > > >
> > > > For the bandit literature, the regret is a critical performance metrics. I suggest the authors to complete the regret analysis for contextual bandits to demonstrate the superiority of the proposed SAU measures on regret performance.
> > > >
> > > > As for the experiments, comparisons with recent deep algorithms are helpful for readers to understand the empirical performance of SAU-based deep algorithms. It would be better if the authors include these experimental results and comparisons in their revision.

---

> > > > > ### Author Response · Authors · 2021-08-19
> > > > > **Thank you for latest comments**
> > > > >
> > > > > Thank you for the additional comments.
> > > > > As we said, we will be happy to add the comparisons with the mentioned deep algorithms, as the reviewers is requesting. As mentioned, in preliminary experiments SAU easily outperforms NeuralTS and NeuralUCB in performance, and in addition is much more scalable and computationally efficient.
> > > > > May we ask whether the reviewer is asking for these additional experiments as a condition for the acceptance score to be increased?
> > > > >
> > > > > As for the theoretical analysis, we'd like to push back a little, and note that the fact that SAU outperforms NeuralTS and NeuralUCB demonstrates how limited the importance of a regret bound is in the contextual setting in practice. NeuralTS and NeuralUCB might come with a regret bound, but first of all, this is only valid in the very restrictive NTK regime (where the interest of a neural network model is arguably limited). Secondly, a theoretical regret bound doesn't guarantee that the algorithm will be beaten by another algorithm (in fact NeuralSAU-Sampling outperformns NeuralTS and NeuralUCB in the very benchmarks used in the NeuralTS paper). Third, NeuralTS and NeuralUCB are impossible to scale up to realistic large-scale datasets and value models of any practical interest, which makes their theoretical properties even less relevant.
> > > > > SAU has solid regret bounds in the multi-armed bandits setting, and other than that it has great potential for practical applications, owning to its scalability, ease of implementation and out-of-the-box compatibility with neural network value models.
> > > > > Based on these points, we'd like to ask the reviewer to kindly reconsider the relative importance of these factors, over the requested theoretical bound.

---

> > > > > > ### Comment · Reviewer_bpP1 · 2021-08-21
> > > > > > **Response to the Authors**
> > > > > >
> > > > > > Thank you for your further explanation.
> > > > > >
> > > > > > First, I think the comparisons with NeuralUCB/NeuralTS in experiments are what you should have included in the submission. It is not a condition for the acceptance score.
> > > > > >
> > > > > > Second, I give the score ‘5’ because I think your theoretical analysis is not complete enough. This paper studies 3 settings, i.e., multi-armed bandit, linear contextual bandit and deep contextual bandit settings, but only provides regre analysis for the first preliminary multi-armed bandit setting. In contrast, most of prior works on linear contextual bandits [Abbasi-Yadkori et al. 2011] and deep contextual bandits [Zhou et al. 2020, Zhang et al. 2021] provide regret analysis. Without regret guarantees, it is hard to determine the efficacy of the SAU measure in these two settings.
> > > > > > While your analysis for the multi-armed bandit setting is interesting and solid, I am more interested in the analysis for contextual bandits, since contextual bandit is a more general setting that can be combined with neural networks naturally and applied in many real-world tasks, e.g., personalized recommendation systems.
> > > > > >
> > > > > > The idea of SAU is very interesting. I sincerely suggest you to further improve the theoretical analysis and submit to top venues.

---

### Official Review · Reviewer_TBN4 · 2021-07-13

**Rating:** 5
**Confidence:** 4

**Summary:**

This paper proposes a new uncertainty measure for contextual bandits named Sample Average Uncertainty (SAU). SAU is derived from a frequentist point of view by directly looking at statistics from model predictions/outputs, and two SAU-based exploration schemes are proposed accordingly (similar to UCB and TS). The simplicity of SAU makes it scalable and compatible with complex function approximators such as neural networks. Theoretically, the authors showed that SAU asymptotically approximates TS for Bernoulli multi-armed bandits. Empirically, SAU-based exploration outperformed TS in both linear and neural contextual bandits.

**Limitations And Societal Impact:**

The authors have discussed limitations and potential negative social impact of their work. However, mitigation strategies for negative impact were not fully discussed.

**Main Review:**

Although a lot of effort has been made in the literature to learn and approximate Bayesian posterior, this paper proposes a simple yet effective frequentist approach for uncertainty estimate and efficient exploration, which does not require maintaining a posterior distribution. The approach is tractable and scales well for problems that require flexible generalization with neural networks. Claims in the paper are appropriately supported by theoretical analysis and empirical results, and the overall paper is well-written and well-organized.

The experiments show that the SAU-based exploration is quite promising for contextual bandits. However, my major concerns with the current version are (a) the baselines did not include more recent developments in the related field and (b) methods have not been compared on environments that require sophisticated exploration. Please see detailed comments below.

(a)	Baselines: Hypermodels [1] and NeuralUCB [2] are recently proposed methods for efficient exploration in bandits, which have not been discussed in the current paper. I wonder how well SAU would perform compared to, for example, linear-hypermodel [1] or NeuralUCB [2]. Computational comparison is also welcomed.

(b)	Environment(s): I am interested to see how well SAU can perform against baselines on environments that require efficient exploration. The WHEEL bandit problem introduced in [3] can be a good example. With different difficulty settings $\delta$, we can better gauge the effectiveness and limit of SAU-based exploration.

(c)	Sensitivity analysis: if mini-batches are used for model training, does the performance vary when the number of batches changes? Does the size of the data buffer matter? Finite buffer vs. infinite buffer (that includes all observed data)?

(d)	Proposition 3 (line 160-162): the direction of inequality inside Pr{$\cdot$} seems incorrect. Should be $\geq$ instead of $\leq$? Please check.

(e)	Synthetic datasets: in Section 4.3, SAU-UCB seems to be consistently better than SAU- Sampling. Any thoughts on why this might be the case? What would happen if both K and p keep increasing? e.g. K = 20, p = 100

I am willing to adjust my evaluation should the above issues be addressed and fully discussed.

Ref.:

[1] Dwaracherla, Vikranth, Xiuyuan Lu, Morteza Ibrahimi, Ian Osband, Zheng Wen, and Benjamin Van Roy. "Hypermodels for exploration." arXiv preprint arXiv:2006.07464 (2020).

[2] Zhou, Dongruo, Lihong Li, and Quanquan Gu. "Neural contextual bandits with ucb-based exploration." In International Conference on Machine Learning, pp. 11492-11502. PMLR, 2020.

[3] Riquelme, Carlos, George Tucker, and Jasper Snoek. "Deep bayesian bandits showdown." In International conference on learning representations. 2018.



**Time Spent Reviewing:**

3.5

---

> ### Author Response · Authors · 2021-08-10
> **Response to Reviewer TBN4**
>
> We would like to thank the reviewer for the careful read of our manuscript and the thoughtful comments that we address below.
>
> **(a)** Thanks for pointing out these recent references. We plan to cite them in the revised version of the paper along with a comparison with our algorithm. What we'd like to emphasize is that the main advantage that our proposed algorithm has over the methods developed in these references is the implementation simplicity of SAU. Our SAU algorithms can be in fact implemented as simple drop-in modules on top of a given neural network architecture, which add exploration capabilities without the need to modify the architecture nor the training procedure, and without the addition of any appreciable memory or computation overhead (for instance Table 2 in Appendix D shows that SAU has an execution speed that is comparable to the simplest exploration strategies like epsilon-greedy).
>     Hypermodels on the other hand have to introduce an additional parametrized model on top of the base model, as well as an additional training cycle.
>     NeuralUCB, an extension of LinUCB for NN models, needs to calculate the outer product of the neural network gradient in order to construct the upper confidence bound. This is a quantity which scales quadratically with the number of parameters of the neural network, and therefore becomes prohibitive even with moderately sized models. Even the some proposed approximation to mitigate this problem still need to perform a costly matrix inversion at each step.
>     As mentioned, compared to these methods SAU enjoys much better scalability owning to its modest computation and memory overhead.
>     In the revised version, we plan to provide this discussion comparing our method to these suggested baselines.
>
> **(b)** Thank you for the suggestion of examining the wheel bandit problem as setting where the need of exploration can be set parametrically. We had time to re-implement this benchmark (although we had to arbitrarily choose parameters like the number of steps, which weren't specified in the paper) and were able to check that SAU-Sampling is marginally but consistently better than the best models reported in Riquelme et al. (NeuralLinear and LinearPosterior). We will be happy to add these results in the revised version of the paper.
>
> **(c)** We checked that the performance of our algorithm is moderately robust to changes of mini-batch size (which we modified by up to 50% without noticing systematic changes) and the size of the memory buffer (which we also reduced by half in some test runs). For the main experiments in the paper we anyway matched these parameters to the corresponding baseline. We also released all code, in order to provide a basis to verify reproducibility and robustness of our results.
>
> **(d)** Yes, the reviewer is correct: it should be ">=" instead of "<=" in line 160-162. Thank you for pointing out this typo, which we will correct in the revised version of the paper.
>
> **(e)** We have some thoughts as to why SAU-UCB seems to be consistently better than SAU-Sampling in the synthetic datasets in Section 4.3 (although that is not always the case in general for other datasets). In UCB, each choice of a specific arm decreases the uncertainty associated with it, while increasing the uncertainty of other arms (by virtue of the log n factor at numerator that is common to the exploration bonus of all arms). This means that the probability of choosing a given arm waxes and wanes throughout the sequence of decisions. Unfortunately, this desirable feature is not present in sampling methods based on TS since the distributions evolve independently across arms.
>    For the case of K = 20, p = 100, we had time to check and found that SAU-based strategies are still better than TS, and SAU-UCB achieves better regret than SAU-Sampling. Although K=20 TSdiag works similarly as TS due to the dimension p=100. We will give a more detailed analysis in the revised version of the paper.

---

> > ### Comment · Reviewer_TBN4 · 2021-08-19
> > **Thank you and additional comments**
> >
> > Thanks very much for the detailed response. I believe the discussion in (a), (c) and (e) can be helpful to add in the revision.
> >
> > For baselines and experiments, I would still suggest the authors (1) add comparative results to more recent baselines, e.g., hyper-models or NeuralUCB/NeuralTS, and (2) add experimental results on the WHEEL environment (potentially with various difficulty settings). I do acknowledge the simplicity of the proposed method and understand the complexity of these baselines, but performance comparison can be essential to better gauge the effectiveness of SAU-based methods empirically. Since the focus of the paper is on efficient exploration, evaluation on environments that require sophisticated exploration is needed in my opinion. WHEEL can be a good example, and experimenting with different difficulty levels is helpful for a thorough understanding of both the efficacy and the limitation of SAU-based methods.

---

### Official Review · Reviewer_R6yL · 2021-07-14

**Rating:** 7
**Confidence:** 2

**Summary:**

This paper introduces a simple exploration strategy called Sample Average Uncertainty (SAU) for contextual bandit problems. It estimates the mean and variance of the reward distribution for different actions, where the estimate of the mean is given by a parametric function that is trained on experience while the estimate of the variance is a simple sample average of the residuals between observed and predicted rewards for each action. It then defines a UCB-like objective and a sampling based objective to transform these predictions into actions, taking into account the exploration/exploitation trade-off. This sampling strategy is analysed theoretically and evaluated empirically, where the results demonstrate the benefit of SAU in terms of achieving lower cumulative regret than baselines.

This is not my main area of expertise and I did not go through all the math in detail, so it is difficult for me to assess the novelty as I am unfamiliar with directly related work. However, it seems to me that the paper presents a simple and useful strategy that is effective in addressing the exploration/exploitation trade-off for contextual bandits. Given the importance of the general exploration/exploitation problem and the simplicity of the approach, this paper may contain more generally useful ideas and therefore my evaluation is that the paper could be accepted.

**Ethical Concerns:**

I do not see any direct ethical concerns.

**Limitations And Societal Impact:**

Obvious limitations are adequately discussed in the discussion section, although I would find it helpful to discuss the implications/limitations of the implicit assumption that the variance does not depend on the context.

**Main Review:**

**Originality**
The paper presents a simple idea but I am not familiar enough with related literature to completely assess the novelty. In this respect, it would be helpful if the paper would include a more structured discussion of related work, which is a bit scattered through the paper and not discussed in much detail.

**Quality**
The proposed methods are principled and well motivated and I cannot spot any obvious flaws. The claims made by the paper are supported by the empirical results. Overall, I think this paper is of good quality.

**Clarity**
The overall idea of the paper was very clear and easy to understand. If the main goal of the paper is to present a practical method, then I think some of the math heavy analysis could be deferred to smooth the overall flow of the paper, but it does not really bother either.  The pseudocodes are very helpful for understanding how to actually implement the proposed strategy and see the relation to e.g. Thompson Sampling.

**Significance**
The exploration-exploitation trade-off is important and the paper presents a simple and effective strategy to address this. As variants of baselines such as UCB are widely used, the ideas presented in this paper may be of interest to a wider audience.

**Questions**
Do I understand correctly that there is an implicit assumption that the variance does not depend on the context. What is the implication of this assumption? Could we (like the mean) also use a parametric estimate of the variance that does depend on context?
Is the term (log n)/n_a in the SAU-UCB necessary? As the variance of the mean already decreases as n_a increases (since tau_a^2 = S_a^2/n_a), can't we just use something like \tilde{u}_n,a = \hat{u}_n,a + 2 \tau (standard confidence interval UB).

**Time Spent Reviewing:**

3

---

> ### Author Response · Authors · 2021-08-10
> **Response to Reviewer R6yL**
>
> Thank you very much for the positive comments expressed on our paper. We're very glad that our presentation was clear, and that the reviewer appreciated the empirical evaluation of our method.
>
> In the revised paper, we plan to include a section discussing the novelty of our approach in the context of the related literature in the field in a more self-contained way, as suggested by the reviewer.
>
> Here below we address the questions of the reviewer.
>
> * **Parametric estimate of the variance that depends on context:**
>
>    As the reviewer correctly points out in the first question, the SAU quantity provided by our algorithm does not indeed depend on the current context (although it does so indirectly, through its dependence on past observed contexts).
>     This can be thought of as an assumption of "noise homogeneity", i.e. the noise variance $\sigma_a^2$ just depends on arm $a$. While seemingly restrictive, we postulate that this assumption might be a good approximate description of the data-generating process underlying complex reward functions such as those described by neural networks, particularly when exploration is important, i.e. during the initial phase of learning when there still wouldn't be anyway enough data to fit a good estimate of noise as a function of context, and it therefore makes sense to assume at most a weak dependence on context. This would explain why SAU works empirically well in the deep learning setting.
>     In addition, noise independence from context could also be seen as a design choice for our algorithm, taken from the perspective of crafting an algorithm that would be as simple and as flexible as possible, and therefore considers a noise variance that only depends on the action, before introducing a dependence on context in its further elaborations.
>     In the future, we in fact think that introducing an explicit dependence of the reward uncertainty on the current context would be a very interesting extension of our work. This could be for instance achieved by explicitly fitting the relation of the SAU measure as a function of context. We would however reserve this type of extensions to future papers, as it would introduce considerable complexity and possibilities in terms of design choices on top of the action-value function.
>     In the revised version of our paper we will definitely clarify this point.
>
> * **Term (log n)/n_a in the SAU-UCB and variance of the mean decreasing as n_a increases:**
>
>     In the paper we use $\tau_a^2$ to denote an approximate estimator of $\bar{\sigma}_a^2$, not an approximate estimator of $var(\bar{r}_a)$, which on the other hand corresponds to $\tau_a^2/{n_a}$. We refer to equation (4) for the details.
>     This is also what explains the factor (log n)/n_a in SAU-UCB.
>
> * **Limitations:**
>
>     We appreciate the suggestion of discussing the independence from context in the limitations section, which we will be happy to do in the revisions. As mentioned, we initially thought of this as a design choice, but it also makes sense to think of it as a limitation that can stimulate improvements and research directions for future publications.

---

> > ### Comment · Reviewer_R6yL · 2021-08-16
> > **Thanks**
> >
> > Thanks for the clarifications.
> >
> > I am still not sure how you would combine SAU with uncertainty based on context as I don't think you can keep a sample average for every context? So this seems a fundamental limitation of using the sample average which is why I suggest discussing it. Or am I missing something?
> >
> > Following the other reviewers suggestions regarding baselines and regret analysis, I strongly encourage the authors to address these.

---

> > > ### Author Response · Authors · 2021-08-17
> > > **Clarification and comparisons to additional baselines**
> > >
> > > Apologies for not being completely explicit, we're happy to clarify.
> > > A context-dependent SAU is actually quite straight-forward to implement (at least in principle). The idea is simply to fit the prediction residuals $e_n^2$ as a function of the context $x_n$. In other words, we fit a regression model $e_n^2=f(x_n, a_n)$ on observed pairs of $e_n^2$'s and $(x_n, a_n)$'s. Such model can for instance be a new neural network, or one that shares some parameters with the value network. When and how much this would help is mostly an empirical question, but in any case it is not a fundamental limitation of our method to be constrained to context-independent models of uncertainty.
> > >
> > > In regard to the additional baselines mentioned by Reviewer TBN4, we started conducting some comparisons with those papers. We're still on the fence on whether these are actually meaningful baselines, since the huge computational cost of these methods does not give them any chance of scaling up to large models and datasets as SAU does, making them inpractical for relevant industrial use cases.
> > > At any rate, for completeness we preliminarily replicated the main empirical results presented in the NeuralTS paper (which reports also NeuralUCB metrics) on the Mushroom and Shuttle datasets (as used in the NeuralTS paper) and found that Neural-SAU-Sampling on average outperforms both NeuralTS and NeuralUCB by more than 50% in terms of final cumulative regret (for the number of steps that they ran the bandits over). We'll be happy to discuss these comparisons more in details in the revised version of the paper, if the reviewers deem that important to get the paper accepted.

---

### Decision · Program_Chairs · 2021-09-28

**Decision:**

Accept (Poster)

**Comment:**

Two reviewers recommended acceptance of the paper (1x weak accept, 1xaccept) and two reviewers recommended (weak) rejection of the paper. The most positive review had low confidence and I downweighed it when comming to a final decision for the paper. The reviewers acknowledged the simple and efficient proposed exploration approach but also raised concerns regarding completeness of the theoretical analysis and the presented empirical evaluation. While a more complete theoretical analysis was not deemed crucial for accpetance of the paper, the missing baselines make it impossible to fully (empirically) understand the value of the proposed approached. The authors commented on this issue but did not provide an empirical comparison but rather highlighted (only) the theoretical differences. I am therefore recommending rejection of the paper but at the same time would like to encourage the authors to improve their paper according to the reviewers' suggestions and in particular work on the empirical comparison of the proposed approach to recently proposed algorithms in the field.

**Consistency Experiment:**

NeurIPS has a long history of experimentation. In 2014, NeurIPS ran an experiment in which 10% of submissions were reviewed by two independent committees to quantify the randomness in the review process. This year, we repeated a variant of this experiment to see how the quality of the review process has changed over time.  This paper was part of the experiment and was therefore assigned to two committees (consisting of reviewers, an Area Chair, and a Senior Area Chair) that reached independent decisions.  If both committees made the same recommendation, this recommendation was followed. If a single committee recommended acceptance, the paper was accepted (with the exception of a few cases in which the other committee identified what we considered a fatal flaw, e.g., an error in a key result).

This copy’s committee reached the following decision: **Reject**

The other committee assigned to the paper recommended **Accept (Poster)**.  You can find the other set of reviews, along with any follow up discussion with the authors here:
https://openreview.net/forum?id=j6TyzaN_P4z